# Diversified Outlier Exposure for Out-of-Distribution Detection via Informative Extrapolation

Jianing Zhu[1]    Geng Yu[2]    Jiangchao Yao[2,3]    Tongliang Liu[4]
Gang Niu[5]    Masashi Sugiyama[5,6]    Bo Han[1,5†]

[1]Hong Kong Baptist University    [2]CMIC, Shanghai Jiao Tong University
[3]Shanghai AI Laboratory    [4]Sydney AI Centre, The University of Sydney
[5]RIKEN Center for Advanced Intelligence Project    [6]The University of Tokyo

{csjnzhu, bhanml}@comp.hkbu.edu.hk
{warriors30, sunarker}@sjtu.edu.cn    tongliang.liu@sydney.edu.au
gang.niu.ml@gmail.com    sugi@k.u-tokyo.ac.jp

## Abstract

Out-of-distribution (OOD) detection is important for deploying reliable machine learning models on real-world applications. Recent advances in outlier exposure have shown promising results on OOD detection via fine-tuning model with informatively sampled auxiliary outliers. However, previous methods assume that the collected outliers can be sufficiently large and representative to cover the boundary between ID and OOD data, which might be impractical and challenging. In this work, we propose a novel framework, namely, *Diversified Outlier Exposure (DivOE)*, for effective OOD detection via informative extrapolation based on the given auxiliary outliers. Specifically, DivOE introduces a new learning objective, which diversifies the auxiliary distribution by explicitly synthesizing more informative outliers for extrapolation during training. It leverages a multi-step optimization method to generate novel outliers beyond the original ones, which is compatible with many variants of outlier exposure. Extensive experiments and analyses have been conducted to characterize and demonstrate the effectiveness of the proposed DivOE. The code is publicly available at: `https://github.com/tmlr-group/DivOE`.

## 1 Introduction

Out-of-distribution (OOD) detection [Nguyen et al., 2015, Hendrycks and Gimpel, 2017] gains increasing attention in deploying machine learning models into open-world scenarios, as deep learning systems are expected to conduct reliable predictions on in-distribution (ID) data while identifying OOD inputs [Bommasani et al., 2021, Hendrycks et al., 2022]. It becomes more critical in safety-critical applications like financial or medical intelligence, where the false prediction for samples out of pre-defined label space can sometimes be a disaster. Extensive explorations [Hendrycks et al., 2019, Liu et al., 2020, Tack et al., 2020, Mohseni et al., 2020, Sehwag et al., 2021, Yang et al., 2021, 2022] in recent years has been contributed to improving the model ability for OOD detection.

Compared with post-hoc methods [Hendrycks and Gimpel, 2017, Liang et al., 2018, Liu et al., 2020, Huang et al., 2021, Sun et al., 2022] which designs different score functions for OOD uncertainty estimation, Outlier Exposure (OE) [Hendrycks et al., 2019] is another kind of method owning better performance improvement on OOD detection [Mohseni et al., 2020, Sehwag et al., 2021, Ming et al., 2022, Katz-Samuels et al., 2022, Wang et al., 2023]. The latter engages surrogate OOD data during

---

[†]Correspondence to Bo Han (bhanml@comp.hkbu.edu.hk).

37th Conference on Neural Information Processing Systems (NeurIPS 2023).

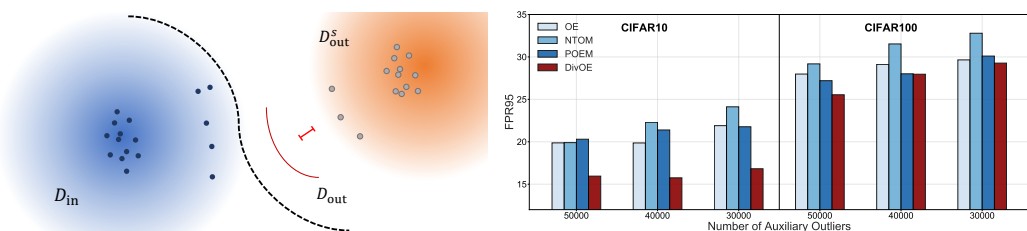

Figure 1: Illustration about the research problem on auxiliary outliers studied in our work for OE. Left panel: the general assumption on the auxiliary outliers, in which the surrogate OOD distribution $\mathcal{D}^s_{\text{out}}$ can not always be broad enough to well represent the unseen OOD distribution $\mathcal{D}_{\text{out}}$ since it is impractical or even infeasible to accurately pre-define and collect those boundary outliers (especially those close to $\mathcal{D}_{\text{in}}$); Right panel: an empirical verification on OOD detection performance (evaluated by FPR95, the lower is the better) with the gradually decreased number of given auxiliary outliers. Our proposed DivOE can perform better via extrapolation than previous sampling-based methods, given different numbers of auxiliary outliers. More experimental details are provided in Appendix C.

training and regularizes the model via fine-tuning those auxiliary outliers. The conceptual idea is to learn the knowledge from auxiliary outliers for effectively identifying OOD inputs. Under such a learning framework, there is an intuitive gap between the surrogate OOD data and the unseen OOD inputs [Fang et al., 2022, Ming et al., 2022] as illustrated in Figure 1. It will be a fundamental problem for boosting the model's discriminative capability on the OOD inputs. Given the finite auxiliary outliers, it naturally motivates the following critical research question: *How could we utilize the given outliers for effective OOD detection if the auxiliary outliers are not informative enough*?

Addressing the essential distribution gap between surrogate OOD data and the unseen OOD inputs remains challenging [Hendrycks et al., 2019, Fang et al., 2022, Du et al., 2022, Yang et al., 2021], as it is hard to know the prior knowledge of potential OOD inputs would be encountered at inference stage [Hendrycks et al., 2022], and intentionally collect them. As the unseen OOD inputs in the real-world scenarios are complex [Cimpoi et al., 2014, Yu et al., 2015, Yang et al., 2022], the performance of OE-based methods is heavily affected by the given auxiliary outliers (as empirically presented in Figure 2). Given the finite auxiliary outliers, the surrogate OOD data are expected to have less discrepancy [Hendrycks et al., 2019, Fang et al., 2022] with the unseen OOD inputs, especially for the ones that are close to the decision boundary. Therefore, a potential idea is to expand the given outliers to cover more informative distributions and better generalize to the unseen OOD inputs.

Based on the previous analysis, we propose a new learning framework, i.e., Divsified Outlier Exposure (DivOE), to alleviate the pessimism of limited informative auxiliary outliers. At the high level, we aim to diversify the current distribution represented by the surrogate OOD data (e.g., Figure 2). In detail, we introduce a novel learning objective (i.e., Eq. (4)) that conducts informative extrapolation based on the given auxiliary outliers. Through a multi-step optimization target for generating a fraction of more informative outliers, DivOE can adaptively extrapolate and learn beyond the original OOD distribution. It realizes expanding the overall surrogate distribution of OOD data toward a broader coverage by maximizing the difference between the extrapolated one with its original counterpart.

We conduct extensive experiments to characterize and present our proposed methods. We have verified its effectiveness with a series of OOD detection benchmarks mainly on two common ID datasets, i.e., CIFAR-10 and CIFAR-100, and also demonstrated its scalability using the ImageNet dataset with the large-scaled auxiliary outliers sampled from ImageNet21K. Under the various evaluations, our DivOE via informative extrapolation, can obtain the better OOD discriminative capability for the models and consistently achieve the lower averaged FPR95 compared with different baselines. Finally, a range of ablation studies from various aspects of the learning framework and further discussions from different perspectives are provided. Our main contributions are summarized as follows,

- Conceptually, we study a more general and practical research setting in outlier exposure for OOD detection, considering the auxiliary outliers having limited information that can not well represent the whole decision areas for ID and OOD data. (in Section 3.1)

- Technically, we propose a novel learning framework, namely *Diversified Outlier Exposure* (DivOE), for facilitating fine-tuning with auxiliary outliers, which conducts informative extrapolation to adaptively diversify outlier exposure. (in Sections 3.2 and 3.3)

- Empirically, extensive explorations from different perspectives are conducted to verify the effectiveness of DivOE in improving OOD detection performance, and we perform various ablations or further discussions to provide a thorough understanding. (in Section 4)

## 2 Background

In this section, we briefly introduce the preliminaries and related work in OOD detection.

### 2.1 Preliminaries

We consider multi-class classification as the original training task [Nguyen et al., 2015], where $\mathcal{X} \subset \mathbb{R}^d$ denotes the input space and $\mathcal{Y} = \{1, \ldots, C\}$ denotes the label space. A reliable classifier should be able to figure out the OOD input, which can be considered a binary classification problem. Given $\mathcal{P}$, the distribution over $\mathcal{X} \times \mathcal{Y}$, we consider $\mathcal{D}_{\text{in}}$ as the marginal distribution of $\mathcal{P}$ for $\mathcal{X}$, namely, the distribution of ID data. At test time, the environment can present a distribution $\mathcal{D}_{\text{out}}$ over $\mathcal{X}$ of OOD data. In general, the OOD distribution $\mathcal{D}_{\text{out}}$ is defined as an irrelevant distribution of which the label set has no intersection with $\mathcal{Y}$ and thus should not be predicted by the model $f(\cdot)$. A decision model $g(\cdot)$ can be made with the threshold $\lambda$:

$$g_\lambda(x; f) = \begin{cases} \text{ID} & S(x; f) \geq \lambda \\ \text{OOD} & S(x; f) < \lambda \end{cases}, \tag{1}$$

Building upon the model $f \in \mathcal{H} : \mathcal{X} \to \mathbb{R}^c$ trained on ID data with the logit outputs, the goal of decision is to utilize the scoring function $S : \mathcal{X} \to \mathbb{R}$ to distinguish the inputs of $\mathcal{D}_{\text{in}}$ from that of $\mathcal{D}_{\text{out}}$ by $S(x; f)$. If the score value is larger than the threshold $\lambda$, the associated input $x$ is classified as ID and vice versa. We consider several representative scoring functions designed for OOD detection in our exploration, e.g., MSP [Hendrycks and Gimpel, 2017], ODIN [Liang et al., 2018], and Energy [Liu et al., 2020]. More detailed definitions are provided in Appendix A.

To mitigate the issue of over-confident predictions for [Hendrycks and Gimpel, 2017, Liu et al., 2020] some OOD data, another line of research directions [Hendrycks et al., 2019, Tack et al., 2020] utilize the auxiliary unlabeled dataset to regularize the model behavior. Among them, one representative baseline is outlier exposure (OE) [Hendrycks et al., 2019]. OE can further improve the detection performance by making the model $f(\cdot)$ finetuned from a surrogate OOD distribution $\mathcal{D}_{\text{out}}^s$, and its corresponding learning objective is defined as follows,

$$\mathcal{L}_{\text{OE}} = \mathbb{E}_{\mathcal{D}_{\text{in}}} \left[ \ell_{\text{CE}}(f(x), y) \right] + \lambda \mathbb{E}_{\mathcal{D}_{\text{out}}^s} \left[ \ell_{\text{OE}}(f(x)) \right], \tag{2}$$

where $\lambda$ is the balancing parameter, $\ell_{\text{CE}}(\cdot)$ is the Cross-Entropy (CE) loss, and $\ell_{\text{OE}}(\cdot)$ is the Kullback-Leibler divergence to the uniform distribution, which can be written as $\ell_{\text{OE}}(h(\boldsymbol{x})) = -\sum_k \texttt{softmax}_k f(x)/C$, where $\texttt{softmax}_k(\cdot)$ denotes the $k$-th element of a softmax output. The OE loss $\ell_{\text{OE}}(\cdot)$ is designed for model regularization, encouraging the model to learn knowledge from the surrogate OOD inputs and return low-confident predictions [Hendrycks and Gimpel, 2017].

### 2.2 Related Work

**OOD Detection.** [Hendrycks and Gimpel, 2017] formally benchmarks the OOD detection problem, proposing to use softmax prediction probability as a conventional baseline method. Subsequent works [Sun et al., 2021] keep focusing on designing post-hoc metrics to distinguish ID samples from OOD samples, among which ODIN [Liang et al., 2018] introduces small perturbations into input images to facilitate the separation of softmax score, Mahalanobis distance-based confidence score [Lee et al., 2018b] exploits the feature space by obtaining conditional Gaussian distributions, energy-based score [Liu et al., 2020] aligns better with the probability density. Except directly designing new score functions, some works [Lin et al., 2021, Sun et al., 2022, Djurisic et al., 2023] pay attention to various aspects to enhance the OOD detection such that LogitNorm [Wei et al., 2022] produces confidence scores by training with a constant vector norm on the logits, DICE [Sun and Li, 2022] reduces the variance of the output distribution by leveraging the model sparsification, and ASH [Djurisic et al., 2023] explores the manipulation of feature representation to enhance OOD detection. In addition, Zhu et al. [2023] improves detection performance by investigating the quality of ID data.

**OOD Detection with Auxiliary Outliers.** Another promising direction toward OOD detection involves the auxiliary outliers for model regularization. From the data perspective, some works

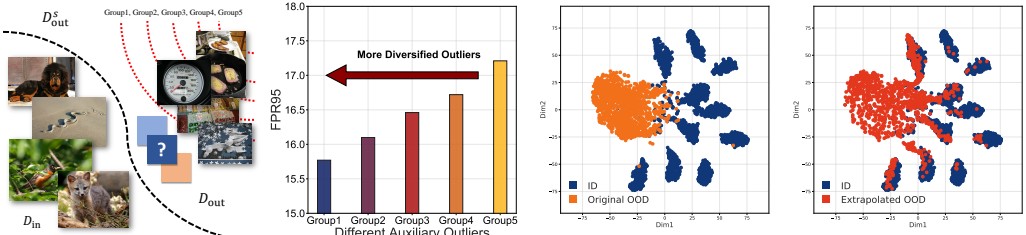

Figure 2: Empirical demonstration about informative outliers in OE and the diversification effect of the proposed DivOE. In the two left panels, we illustrate auxiliary outliers in different informative levels and divide them into five groups according to the degree of diversity. Empirically, we show that the OOD detection performance of OE can be benefited by the most diversified Group1 among these trials. We leave the experimental details in Appendix C for reference. In the two right panels, we compare the TSNE visualization of the original outliers with our extrapolated outliers in DivOE.

explore generating virtual outliers such that VOS [Du et al., 2022] and NPOS [Tao et al., 2023] regularize the decision boundary by adaptively sampling virtual outliers from the low-likelihood region. On the other hand, other works exploit information from natural outliers, such that outlier exposure is introduced by [Hendrycks et al., 2019], given that real OOD data are available in enormous quantities. Recently, some works highlight the importance of sampling strategy, such that NTOM [Chen et al., 2021] greedily utilizes informative auxiliary data to tighten the decision boundary for OOD detection, and POEM [Ming et al., 2022] adopts Thompson sampling to contour the decision boundary precisely. The performance of training with outliers is usually superior to that without outliers, as shown in many other works [Fort et al., 2021, Salehi et al., 2021, Sun et al., 2023]. From the optimization perspective, Wang et al. [2023] introduces implicit regularization via model perturbation, and Wang et al. [2022], Yang et al. [2023] present early trials to improve OOD performance in terms of the AUROC with tighter generalization bounds and new evaluation metrics.

## 3 Method: Diversified Outiler Exposure

In this section, we introduce our new framework, i.e., *Diversified Outlier Exposure* (DivOE), which conducts informative extrapolation during fine-tuning with auxiliary outliers. First, we present and discuss the critical motivation that inspires our method (Section 3.1). Second, we introduce its newly derived learning objective and explain the underlying implications (Section 3.2). Lastly, we present the algorithmic realization of DivOE and discuss its compatibility with other methods (Section 3.3).

### 3.1 Motivation

First, collecting OOD samples near the boundary still remains challenging as we can hardly know which kind of samples are truly located in the decision boundary between ID and OOD space. Second, as empirically shown in the left-middle panel of Figure 2, if the auxiliary outliers are not sufficiently informative (e.g., less diversified), the performance of outlier exposure will be limited due to the under-represented OOD distributions, which can be reflected by the higher FPR95 score (indicating a higher error on OOD detection). Thus, it naturally arises the following research question,

> *How could we utilize the given outliers for effective OOD detection if the auxiliary outliers are not informative enough to represent the unseen OOD distribution?*

Especially for those OOD inputs that are close to the decision boundary, the auxiliary outliers may not well characterize such a broad distributional area [Fang et al., 2022, Wang et al., 2023, Zhang et al., 2023]. Therefore, a new mechanism is required to diversify the outlier exposure by exploring more potential OOD distribution for effective OOD detection. As in the right two panels of Figure 2, DivOE realizes this expectation by explicitly extrapolating the OOD data engaged during training.

### 3.2 DivOE via Informative Extrapolation

As aforementioned, the OE paradigm heavily relies on the auxiliary outliers sampled from the surrogate OOD distribution. Given the auxiliary outliers that are not informative enough to represent the

unseen OOD data, the model is expected to learn beyond the current surrogate OOD distribution. One conceptual idea to achieve this goal is to extrapolate based on the current surrogate distribution. Under such a concurrent learning paradigm, the model can be regularized by the original surrogate OOD distribution and simultaneously generalize beyond the given auxiliary outliers for OOD detection. To this intuition, we consider an adaptively evolved version of the surrogate OOD distribution $\mathcal{D}_{\text{out}}^{s}$ under the learning framework of OE as follows,

$$\mathcal{L}_{\text{OE}}^{*} = \mathbb{E}_{\mathcal{D}_{\text{in}}}\left[\ell_{\text{CE}}(f(x), y)\right] + \lambda \mathbb{E}_{\mathcal{D}_{\text{out}}^{e}}\left[\ell_{\text{OE}}(f(\tilde{X}))\right], \tag{3}$$

where $\mathcal{D}_{\text{out}}^{e}$ indicates the extrapolated surrogate distribution during training and $\tilde{X}$ indicates the newly manipulated samples that are different from those in the original objective (i.e., Eq. (2)) of OE. In the above learning objective, the left part is for the original classification task, and the right part is for the newly proposed informative extrapolation in our DivOE. With the conceptual target for diversifying the current surrogate OOD distribution, we consider reformulating the specific objective as,

$$\mathbb{E}_{\mathcal{D}_{\text{out}}^{e}}\left[\ell_{\text{OE}}(f(\tilde{X}))\right] = (1-\beta)\ell_{\text{OE}}(f; \mathcal{D}_{\text{out}}^{s}) + \beta \underbrace{\max_{\Delta}\left[\ell_{\text{OE}}(f; \mathcal{D}_{\text{out}}^{s+\Delta}) - \ell_{\text{OE}}(f; \mathcal{D}_{\text{out}}^{s})\right]}_{\text{Informative Extrapolation}}, \tag{4}$$

where $\beta$ indicates a balancing factor for controlling the extrapolation ratio and $\mathcal{D}_{\text{out}}^{s+\Delta}$ represents the synthesized distribution with a manipulation $\Delta$. Intuitively, the overall objective defined in Eq. (4) not only learns from the original surrogate OOD distribution but also generalizes to the different OOD distributions via the maximization part. The underlying insight is to expand the surrogate OOD distribution towards a more diversified one. The informative extrapolation is realized by the maximization part that maximizes the differences between the original surrogate OOD distribution and the generated one. Note that, Eq. (4) corresponds to an expectation formulation based on the whole population for diversifying outlier exposure, which is prohibitively expensive in realization. To achieve that more efficiently, we introduce an instance-level version for practical realization, and its optimization target can be formulated as the following hybrid loss,

$$\ell_{\text{OE}}(f(\tilde{X})) = \frac{1}{n-rn}\underbrace{\sum_{n-rn}\ell_{\text{OE}}(f(x))}_{\text{Original Distribution}} + \frac{1}{rn}\underbrace{\sum_{rn}\max_{\delta}\ell_{\text{OE}}(f(x+\delta))}_{\text{Informative Extrapolation}}, \tag{5}$$

where $n$ indicates the sample number of auxiliary outliers and $r$ indicates the extrapolation ratio that corresponds to $\beta$ in Eq. (4). Except for the original loss on learning with the given outliers, the latter part of Eq. (5) defines a surrogate optimization part for realizing the informative extrapolation.

Here we adopt instance-level maximization for manipulating the expected OOD samples based on the original one, in which the specific $\delta$ is obtained by the following optimization objective,

$$\delta = \arg\max_{\delta}\ell_{\text{OE}}(f(x+\delta)), \tag{6}$$

Intuitively, DivOE will generate new outliers closer to the decision boundary to expand the coverage area of auxiliary outliers, and simultaneously be more informative to shape the decision area between ID and OOD data. The specific optimization process of $\delta$ is characterized as,

$$\delta \leftarrow \delta + \alpha \text{sign}(\nabla_{\delta}\ell_{\text{OE}}(x+\delta)), \quad \text{sign}(\nabla_{\delta}\ell_{\text{OE}}(x+\delta)) \leftarrow \text{sign}(\nabla_{\delta}\ell_{\text{OE}}^{S(\cdot;f)}(x+\delta)), \tag{7}$$

where sign is the function that extracts the sign of the tensor elements and $S(\cdot; f)$ indicates the general OOD score functions. The optimization process also indicates that our extrapolation framework is general and can adopt different score functions to generate targeted outliers. Here we also provide the theoretical implication of our proposed objective based on previous work [Ming et al., 2022].

**Theorem 3.1.** *Given a simple Gaussian mixture model in binary classification with the hypothesis class $\mathcal{H} = sign(\theta^T x), \theta \in \mathbb{R}^d$. There exists constant $\alpha$, and $\sigma(\epsilon)$ that satisfy,*

$$\frac{\mu^T \theta_{n_1,n_2}^*}{\sigma \|\theta_{n_1,n_2}^*\|} \geq \frac{\|\mu\|^2 - \sigma^{\frac{1}{2}}\|\mu\|^{\frac{3}{2}} - \frac{\sigma^2(\alpha-\tau(\epsilon))}{2}}{2\sqrt{\frac{\sigma^2}{n}(d+\frac{1}{\sigma}) + \|\mu\|^2}}. \tag{8}$$

From the perspective of sample complexity, we present the effects of our informative extrapolation. As $\text{FPR}(\theta_{n_1,n_2}^*) = erf(\frac{\mu^T \theta_{n_1,n_2}^*}{\sigma \|\theta_{n_1,n_2}^*\|})$ is monotonically decreasing, the lower bound of $\frac{\mu^T \theta_{n_1,n_2}^*}{\sigma \|\theta_{n_1,n_2}^*\|}$ will

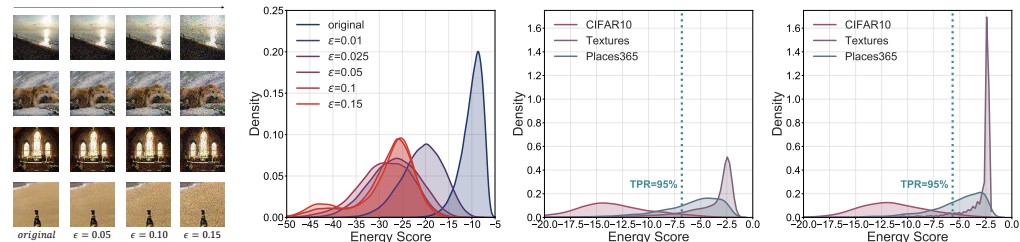

Figure 3: Empirical exploration of informative extrapolation. Left panel: visualization of extrapolated outliers w.r.t different diversified strength $\epsilon \in [0, 0.15]$; Left-middle panel: the corresponding density of the energy score distributions on the different extrapolated outliers; Right-middle panel: comparison of ID/OOD distributions based on Energy score of the OE baseline; Right panel: comparison of ID/OOD distributions based on Energy score using the model fine-tuned by the proposed DivOE.

increase with the constraint of $(\alpha - \tau(\epsilon))$ (which corresponds to our DivOE that extrapolates the outlier boundary towards ID data) decrease in our methods, the upper bound of $\mathrm{FPR}(\theta^*_{n_1,n_2})$ will decrease. The above indicates the benefit of DivOE with the informative extrapolation on shaping boundary decision areas. The completed definition and the analysis can be referred to in Appendix B.

## 3.3 Realization of DivOE

In this part, we introduce the realization of the whole learning framework of DivOE in detail. Here we formally present the learning objective of DivOE via informative extrapolation as follows,

$$\mathcal{L}_{\mathrm{DivOE}} = \frac{1}{m} \sum_m \ell_{\mathrm{CE}}(f(x), y) + \lambda \left( \frac{1}{n - rn} \sum_{n-rn} \ell_{\mathrm{OE}}(f(x)) + \frac{1}{rn} \sum_{rn} \ell_{\mathrm{OE}}(f(x + \delta)) \right), \quad (9)$$

where $m$ and $n$ indicate the sample numbers of ID data and auxiliary outliers respectively, other notations keep the same meaning as the previous equations. Concretely, the maximization part of DivOE adopts the multi-step optimization (e.g., projected gradient decent [Bottou, 2012, Madry et al., 2018]) to realize the targeted outlier synthetics. The specific process is defined as,

$$(x)^{(t+1)} = \Pi_{\mathcal{B}[x^{(0)};\epsilon]} \left( x^{(t)} + \alpha \mathrm{sign}(\nabla_{x^{(t)}} \ell_{\mathrm{OE}}(f(x^{(t)}))) \right), \quad (10)$$

where $t \in \mathbb{N}$, $x^{(t)}$ is the extrapolated OOD data at step $t$, and $\Pi_{\mathcal{B}[x^{(0)};\epsilon]}(\cdot)$ is the projection function that projects the new OOD data back into the $\epsilon$-ball centered at $x^{(0)}$ if necessary. Different from the conceptual idea of constraining the perturbation (for human-imperceptibility) in the literature on adversarial robustness [Madry et al., 2018, Zhang et al., 2019], the projection operation in DivOE provides a quantification way to study the extrapolated distribution and its diversification. As the generation process starts from the given outliers, the semantical class (belongs to ID or OOD) is hard to change, as empirically shown in the left-most panel of Figure 3 (we adopted a very large $\epsilon = 0.15$). More exploration about this perspective will leave to Section 4.3 for further discussion.

Here we summarize the whole procedure of the proposed DivOE in Algorithm 1, which consists of multi-round training iterations. To be specific, on each round of fine-tuning, DivOE first samples a mini-batch with a specific ratio $r \in [0, 1]$ from the auxiliary outliers and conduct multi-step optimization following Eq. (10) to generate the new outliers. With the ID training data, the original outliers, and the extrapolated outliers, DivOE then fine-tune the model with the loss defined in Eq. (9).

In practical realization, for the specific extrapolation strength $\epsilon$, DivOE can also generate the extrapolation pool with multiple $\epsilon$ candidates within the extrapolated sub-batch. The specific definition of extrapolation pool is as follows,

$$\mathcal{O}(\tilde{X}) = \{x^{*k_1}_{\epsilon_1}, \ldots, x^{*k_p}_{\epsilon_p}\}^{\sum_{i=1}^p k_i = rn}_{\epsilon_1, \ldots, \epsilon_p \in [0,1]}, \quad (11)$$

where the extrapolation strength with its corresponding sample number can be flexibly decided in the extrapolated sub-batch with a total sample number equal to $r * n$, which may better facilitate the diversification target. As a primary exploration, we visualize the extrapolated data with its distribution quantified by Energy score [Liu et al., 2020] in the left two panels in Figure 3. Intuitively, even with a large extrapolation strength with human-perceptible pixel noise added in the original outliers, it will not change its classification results in OOD detection. It can represent a more different OOD distribution compared with the given outliers as the distribution shifted gradually to the right.

---

**Algorithm 1** Diversified Outlier Exposure (DivOE) via Informative Extrapolation

---

**Input:** given model: $\theta$, fine-tuning epochs : $T$, training samples of ID data: $x \sim \mathcal{D}_{\text{in}}^{\text{s}}$, training samples of auxiliary outliers: $x^* \sim \mathcal{D}_{\text{out}}^{\text{s}}$, extrapolated ratio of the original outliers: $r$, extrapolate epsilon: $\epsilon$, optimization step number: $k$, extrapolation pool: $\mathcal{O}(\tilde{X})$;
**Output:** fine-tuned model $\theta^T$;

1: **for** epoch $= 1, \ldots, T$ **do**
2:     **for** mini-batch $= 1, \ldots, M$ **do**
3:         Sample a mini-batch $\{(x_i, y_i)\}_{i=1}^m$ from $\mathcal{D}_{\text{in}}$, sample a mini-batch $\{(x_j^*)\}_{j=1}^n$ from $\mathcal{D}_{\text{out}}^s$
4:         Sample a sub-batch with ratio $r$ from the batch of outliers $\{(x_j^*)\}_{j=1}^n$
5:         **for** $k = 1, \ldots, r * n$ (in parallel) **do**
6:             Obtain the extrapolated outliers $\tilde{x}_k^*$ by Eq. (10) that following $\max \ell_{\text{OE}}(f(\tilde{x}_k^*))$
7:             Aggregate all the extrapolated samples with uniform or different $\epsilon$ in $\mathcal{O}(\tilde{X})$
8:         **end for**
9:         $\theta \leftarrow \theta - \eta \nabla_\theta \left\{ \frac{1}{m} \sum \ell_{\text{CE}}(f(x), y) + \lambda(\frac{1}{n-rn} \sum \ell_{\text{OE}}(f(x^*)) + \frac{1}{rn} \sum \ell_{\text{OE}}(f(\tilde{x}^*))) \right\}$
10:     **end for**
11: **end for**

---

**Comparison and Compatibility.** Compared with the conventional OE algorithm [Hendrycks et al., 2019], the critical difference behind DivOE is trying to extrapolate based on the given auxiliary outliers for diversifying outlier exposure. To be specific, it provides a general framework (under the guidance of the learning objective in Eq. (5)) for modeling the informative unknown inputs beyond the original ones. The discriminative feature regularized by our DivOE can be utilized by those advanced post-hoc scoring functions [Sun et al., 2021, Huang et al., 2021, Sun and Li, 2022, Djurisic et al., 2023]. For OE-based methods, the informative extrapolation introduced in DivOE is orthogonal to current tuning objectives [Hendrycks and Gimpel, 2017, Wang et al., 2023] and also compatible to those different data augmentation [Zhang et al., 2018, Yun et al., 2019], synthesizing [Kong and Ramanan, 2021, Lee et al., 2018a], and also sampling strategies [Chen et al., 2021, Ming et al., 2022] adopted based on either ID data [Du et al., 2022, Tao et al., 2023] or auxiliary outliers.

## 4 Experiments

In this section, we present the comprehensive verification of the proposed DivOE in the OOD detection scenario. First, we provide the experimental setups in detail (in Section 4.1). Second, we provide the performance comparison of our DivOE with a series of post-hoc scoring functions and the OE-based methods with different strategies on the auxiliary outliers (in Section 4.2). Third, we conduct various ablation studies and further discussions to understand our DivOE (in Section 4.3).

### 4.1 Setups

Here we present several critical parts of experimental setups and leave more details in Appendix C.

**Datasets.** Following the common benchmarks used in previous work [Liu et al., 2020, Ming et al., 2022], we adopt CIFAR-10, CIFAR-100 [Krizhevsky, 2009] as our ID datasets. We use a series of different image datasets as the OOD datasets, namely Textures [Cimpoi et al., 2014], Places365 [Zhou et al., 2017], iSUN [Xu et al., 2015], LSUN_Crop [Yu et al., 2015], and LSUN_Resize [Yu et al., 2015]. For the surrogate OOD data used in OE-based methods [Hendrycks et al., 2019, Chen et al., 2021, Ming et al., 2022, Zhang et al., 2023], we adopt Tiny-ImageNet [Le and Yang, 2015]. We also conduct the experiments on the ImageNet, where we utilize the large-scaled ImageNet21K [Ridnik et al., 2021] as the surrogate OOD set like previous works [Wang et al., 2023] for OE-based methods.

**Evaluation metrics.** We employ the following three common metrics to evaluate the performance of OOD detection: (i) Area Under the Receiver Operating Characteristic curve (AUROC) [Davis and Goadrich, 2006] can be interpreted as the probability for a positive sample to have a higher discriminating score than a negative sample [Fawcett, 2006]; (ii) Area Under the Precision-Recall curve (AUPR) [Manning and Schütze, 1999] is an ideal metric to adjust the extreme difference

Table 1: Main Results (%). Comparison with competitive OOD detection baselines. All methods are trained on the same backbone. Values are percentages. Bold numbers are superior results. ↑ indicates larger values are better, and ↓ indicates smaller values are better. (averaged by multiple trials)

| $\mathcal{D}_{in}$ | Method | FPR95↓ | AUROC↑ | AUPR↑ | ID-ACC↑ | w./w.o $\mathcal{D}_{aux}$ | Operation for $\mathcal{D}_{aux}$ |
|---|---|---|---|---|---|---|---|
| **CIFAR-10** | MSP [Hendrycks and Gimpel, 2017] | 52.21 | 90.61 | 97.84 | 94.84 | | |
| | ODIN [Liang et al., 2018] | 32.84 | 90.67 | 97.29 | 94.84 | | |
| | Mahalanobis [Lee et al., 2018b] | 36.27 | 92.69 | 98.31 | 94.84 | | |
| | Energy [Liu et al., 2020] | 32.82 | 91.99 | 97.86 | 94.84 | | |
| | ASH [Djurisic et al., 2023] | 33.01 | 92.11 | 97.92 | 94.84 | | |
| | OE [Hendrycks et al., 2019] | 13.76±0.23 | 97.53±0.03 | 99.43±0.01 | 94.51±0.06 | ✓ | Vanilla |
| | Energy (w. $\mathcal{D}_{aux}$) [Liu et al., 2020] | 18.15±0.32 | 90.90±0.18 | 96.34±0.10 | 94.51±0.01 | ✓ | Vanilla |
| | VOS [Du et al., 2022] | 34.30±0.02 | 91.27±0.01 | 97.64±0.01 | 94.02±0.01 | ✓ | Virtually Synthesized |
| | NTOM [Chen et al., 2021] | 17.76±0.84 | 91.08±0.44 | 96.40±0.18 | **95.04±0.03** | ✓ | Greedy Sampled |
| | POEM [Ming et al., 2022] | 17.81±0.78 | 90.96±0.37 | 96.36±0.15 | 94.90±0.04 | ✓ | Thompson Sampled |
| | MixOE [Zhang et al., 2023] | 13.57±0.25 | 97.47±0.07 | 99.39±0.02 | 94.98±0.02 | ✓ | Mixup with ID Data |
| | **DivOE (Ours)** | **11.66±0.04** | **97.82±0.04** | **99.48±0.01** | 94.66±0.12 | ✓ | **Extrapolated** |
| **CIFAR-100** | MSP [Hendrycks and Gimpel, 2017] | 79.71 | 76.13 | 94.07 | 75.95 | | |
| | ODIN [Liang et al., 2018] | 63.29 | 82.44 | 95.49 | 75.95 | | |
| | Mahalanobis [Lee et al., 2018b] | 52.90 | 83.86 | 95.73 | 75.95 | | |
| | Energy [Liu et al., 2020] | 71.93 | 80.27 | 95.00 | 75.95 | | |
| | ASH [Djurisic et al., 2023] | 61.82 | 83.42 | 95.84 | 75.87 | | |
| | OE [Hendrycks et al., 2019] | 27.67±0.43 | 91.89±0.12 | 97.93±0.04 | 75.41±0.10 | ✓ | Vanilla |
| | Energy (w. $\mathcal{D}_{aux}$) [Liu et al., 2020] | 27.86±0.26 | 90.70±0.07 | 97.40±0.03 | 75.03±0.11 | ✓ | Vanilla |
| | VOS [Du et al., 2022] | 70.87±0.00 | 80.00±0.00 | 94.89±0.00 | **76.25±0.00** | ✓ | Virtually Synthesized |
| | NTOM [Chen et al., 2021] | 27.58±0.88 | 91.08±0.34 | 97.52±0.10 | 75.41±0.10 | ✓ | Greedy Sampled |
| | POEM [Ming et al., 2022] | 26.50±0.10 | 91.33±0.06 | 97.58±0.02 | 75.74±0.14 | ✓ | Thompson Sampled |
| | MixOE [Zhang et al., 2023] | 35.26±0.80 | 90.31±0.16 | 97.58±0.04 | 75.74±0.04 | ✓ | Mixup with ID Data |
| | **DivOE (Ours)** | **24.80±0.51** | **92.91±0.11** | **98.22±0.03** | 75.24±0.01 | ✓ | **Extrapolated** |
| **ImageNet** | MSP [Hendrycks and Gimpel, 2017] | 75.23 | 76.67 | 94.35 | 68.72 | | |
| | ODIN [Liang et al., 2018] | 67.61 | 80.36 | 95.15 | 68.72 | | |
| | Mahalanobis [Lee et al., 2018b] | 83.41 | 58.47 | 87.72 | 68.72 | | |
| | Energy [Liu et al., 2020] | 71.44 | 79.52 | 94.98 | 68.72 | | |
| | ASH [Djurisic et al., 2023] | 73.51 | 77.03 | 93.43 | 68.70 | | |
| | OE [Hendrycks et al., 2019] | 61.94±0.03 | 81.58±0.01 | 95.49±0.00 | 75.90±0.00 | ✓ | Vanilla |
| | Energy (w. $\mathcal{D}_{aux}$) [Liu et al., 2020] | 66.94±0.01 | 79.93±0.02 | 94.90±0.00 | 74.19±0.08 | ✓ | Vanilla |
| | VOS [Du et al., 2022] | 74.00±0.02 | 78.97±0.01 | 94.82±0.00 | 76.15±0.02 | ✓ | Virtually Synthesized |
| | NTOM [Chen et al., 2021] | 66.32±0.24 | 80.14±0.08 | 94.95±0.02 | 74.26±0.01 | ✓ | Greedy Sampled |
| | POEM [Ming et al., 2022] | 67.81±0.44 | 79.36±0.24 | 94.73±0.06 | 74.21±0.03 | ✓ | Thompson Sampled |
| | MixOE [Zhang et al., 2023] | 70.12±0.26 | 78.85±0.05 | 94.85±0.02 | **76.18±0.01** | ✓ | Mixup with ID Data |
| | **DivOE (Ours)** | **60.12±0.11** | **81.96±0.00** | **95.59±0.00** | 75.73±0.02 | ✓ | **Extrapolated** |

between positive and negative base rates; (iii) False Positive Rate (FPR) at $95\%$ True Positive Rate (TPR) [Liang et al., 2018] indicates the probability for a negative sample to be misclassified as positive when the true positive rate is at $95\%$. We also include in-distribution testing accuracy (ID-ACC) to reflect the preservation level of the performance for the original classification task on ID data.

**OOD detection baselines.** We compare the proposed method with several competitive baselines in the two directions. Specifically, we adopt Maximum Softmax Probability (MSP) [Hendrycks and Gimpel, 2017], ODIN [Liang et al., 2018], Mahalanobis score [Lee et al., 2018b], and Energy score [Liu et al., 2020] as scoring function baselines; We adopt OE [Hendrycks et al., 2019], Energy-bounded learning [Liu et al., 2020], NTOM[Chen et al., 2021], POEM [Ming et al., 2022] and MixOE [Zhang et al., 2023] as baselines with outliers. For all scoring function methods, we assume the accessibility of well-trained models. For all methods involving outliers, we constrain all major experiments to a fine-tuning scenario, which is more practical in real cases. Different from training a dual-task model at the very beginning, equipping deployed models with OOD detection ability is a much more common circumstance. We leave more definitions and implementations in Appendix A.

### 4.2 Main Results

In this part, we present the major performance comparison with some representative baseline methods for OOD detection to demonstrate the effectiveness of the proposed DivOE. Specifically, we consider several post-hoc methods of scoring functions as the performance reference based on the pre-trained model and some OE-based methods for specific comparison on fine-tuning with auxiliary outliers. Note that for all the experiments here, we utilize the whole auxiliary outliers (following the general setup of OE [Hendrycks et al., 2019, Liu et al., 2020] and ensuring the same number of outliers engaged in each method) instead of the reduced ones adopted in previous figures for illustrations.

In Table 1, we present the overall results using different methods for OOD detection. Since the OE-based methods engage the auxiliary outliers during training, the model will generally gain better

Table 2: OOD detection performance on compatibility experiments. All methods are trained on the same backbone. ↑ indicates larger values are better, and ↓ indicates smaller values are better.

| Method | $\mathcal{D}_{\text{in}}$ | | | | | | | |
|---|---|---|---|---|---|---|---|---|
| | **CIFAR-10** | | | | **CIFAR-100** | | | |
| | FPR95↓ | AUROC↑ | AUPR↑ | ID-ACC↑ | FPR95↓ | AUROC↑ | AUPR↑ | ID-ACC↑ |
| OE | 13.76±0.23 | 97.53±0.03 | 99.43±0.01 | 94.51±0.06 | 27.67±0.43 | 91.89±0.12 | 97.93±0.04 | **75.41±0.10** |
| **DivOE** | **11.66±0.04** | **97.82±0.04** | **99.48±0.01** | **94.66±0.12** | **24.80±0.51** | **92.91±0.11** | **98.22±0.03** | 75.24±0.01 |
| Energy (w. $\mathcal{D}_{\text{aux}}$) | 18.15±0.32 | 90.90±0.18 | 96.34±0.10 | 94.51±0.01 | 27.86±0.26 | 90.70±0.07 | 97.40±0.03 | 75.03±0.11 |
| **Energy (w. $\mathcal{D}_{\text{aux}}$)+DivOE** | **16.03±0.35** | **92.23±0.10** | **96.87±0.03** | **94.53±0.05** | **24.71±0.29** | **92.04±0.19** | **97.82±0.08** | **75.25±0.11** |
| NTOM | 17.76±0.84 | 91.08±0.44 | 96.40±0.18 | 95.04±0.03 | 27.58±0.88 | 91.08±0.34 | 97.52±0.10 | 75.41±0.10 |
| **NTOM+DivOE** | **15.30±0.57** | **92.21±0.57** | **96.73±0.28** | 94.75±0.02 | **25.97±0.74** | **91.62±0.29** | **97.68±0.10** | 75.39±0.15 |
| POEM | 17.81±0.78 | 90.96±0.37 | 96.36±0.15 | 94.90±0.04 | 26.50±0.10 | 91.33±0.06 | 97.58±0.02 | 75.74±0.14 |
| **POEM+DivOE** | **15.22±0.20** | **92.42±0.08** | **96.90±0.03** | 94.66±0.07 | **25.05±0.03** | **91.91±0.07** | **97.77±0.03** | 75.62±0.05 |
| OECC | 17.65±0.06 | 96.68±0.03 | 99.19±0.01 | 94.50±0.07 | 28.51±0.11 | 91.08±0.08 | 97.70±0.03 | **74.95±0.30** |
| **OECC+DivOE** | **14.18±0.04** | **97.26±0.07** | **99.32±0.03** | 94.49±0.15 | **26.13±0.39** | **92.07±0.14** | **97.97±0.05** | 74.74±0.36 |
| DOE | 7.39±0.46 | 98.29±0.08 | 99.58±0.02 | 92.82±0.08 | 30.69±0.93 | 93.13±0.15 | 98.39±0.03 | 71.41±0.38 |
| **DOE+DivOE** | **6.51±0.10** | **98.60±0.01** | **99.65±0.01** | **93.45±0.02** | **20.26±0.20** | **95.41±0.05** | **98.91±0.02** | 71.43±0.60 |

empirical performance on OOD detection, reflected by the evaluation metrics. For those OE-based methods, we provide a specific comparison on their detailed operation for $\mathcal{D}_{\text{aux}}$ (e.g., $\mathcal{D}_{\text{out}}^s$) in the right of Table 1. Since VOS virtually synthesizes the outliers based on the ID data, it doesn't perform better than those using the real collected outliers. Among the rest, NTOM and POEM adopt different strategies for informative sampling, which obtain different levels of detection performance gains across different ID datasets. MixOE that mixups ID and surrogate OOD data does not perform well on some complexed tasks like CIFAR-100 and ImageNet, since the original algorithm is designed for fine-grained OOD detection. Without sacrificing much classification performance (i.e., ID-ACC) on ID data, our DivOE can consistently achieve better OOD detection performance across the three datasets, which verifies the effectiveness of our methods with the newly proposed informative extrapolation for outlier exposure. We also summarize the fine-grained results on OOD detection to Tables 12 and 13 in Appendix C, which provides a better understanding of the improvement.

In Table 2, we report the results of compatibility experiments, in which we compare those OE-based methods with their variants, incorporating our DivOE for informative extrapolation. Specifically, we consider some of the previous OE-based methods and also add two related methods, e.g., OECC [Papadopoulos et al., 2021] and DOE [Wang et al., 2023], in the comparison. Although those methods can have different level of performance enhancement compared with the conventional OE baseline, We can find that our DivOE can consistently help them gain better OOD detection performance across three evaluation metrics, and keep the ID-ACC comparable with the original pre-trained model. As for DOE, it utilizes the model perturbation on implicit outlier synthesizing, which is also orthogonal and compatible with our DivOE for explicit extrapolation. In this table, we can find the plain DivOE can achieve lower FPR95 on CIFAR-100 with DOE, while the performance of DOE is affected by different ID datasets. The above demonstrates the effectiveness and algorithmic robustness of our proposed method. In addition, we also provide further discussion on the differences in Appendix A.1.

### 4.3 Ablation and Further Discussions

In this part, we conduct further explorations to provide a thorough understanding of our DivOE. For the extra results and discussions (e.g., impact and limitations), we leave more details in Appendix C.

**Importance of extrapolation ratio in diversifying the distribution.** As a critical aspect of our proposed learning objective of DivOE in Eq. (9), the extrapolation ratio is introduced to control the manipulated portion of the given auxiliary outliers. DivOE aims to extrapolate based on the surrogate OOD distribution without total loss of the original representation. In Figure 4(a), we show the performance by varying the ratio $r$. It is worth noting that adopting extrapolation on the full batch of data may even degrade the performance, indicating the vital role of the extrapolation ratio in the whole learning framework. It performs diversifying as the maximization part introduced in Eq. (5) instead of an overall augmentation. The results verify the rationality of pursuing diversified outliers instead of only most informative ones. We provide more comparison on the concepts of "diversified" and "informative" in Appendix A.2, and leave more results in Appendix C to further explain it.

**Extending the diversified strength in the informative extrapolation.** To better characterize the effects of informative extrapolation in DivOE, we consider fine-tuning the given model with

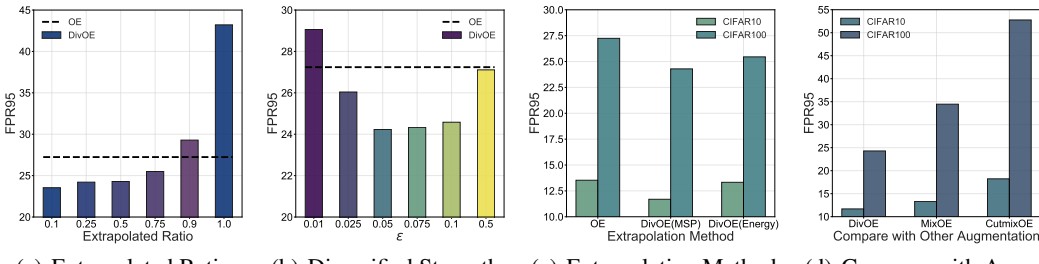

| (a) Extrapolated Ratio | (b) Diversified Strength | (c) Extrapolation Method | (d) Compare with Augs. |

Figure 4: Ablation Study. (a) performance of using different extrapolated ratios of DivOE; (b) exploration of different diversified strength $\epsilon$; (c) using different OOD scores as informative extrapolation targets; (d) comparison between DivOE and OE adopting other augmentations for diversification.

different $\epsilon$ (with a consistent extrapolation ratio, $r = 0.5$), which is used to constrain the diversified strength in maximizing the optimization target in Eq. (6). In Figure 4(b), we can observe that an appropriate diversified strength can help OE performs better on OOD detection, while using a large $\epsilon$ may empirically return to the original baseline case. One possible reason may be that the extrapolated outliers with a large $\epsilon$ may fail to become more informative as previously reflected in Figure 3.

**Generality of using different OOD scores.** Since DivOE introduces a general learning framework of outlier exposure for model finetuning with auxiliary outliers, the specific realization for informative extrapolation can have multiple choices. For example, different scoring functions can all perform OOD uncertainty estimation, indicating the multiple candidate target for Eq. (6) to optimize. Here we report the performance using different optimization targets (e.g., MSP or Energy score) in Figure 4(c), where they have different performance improvements compared with the original OE baseline.

**Comparison with different augmentation-based methods for OE.** In Figure 4(d), we perform the comparison of our DivOE with the original OE adopting some representative augmentation methods (e.g., Mixup [Zhang et al., 2018] and CutMix [Yun et al., 2019]) for diversifying the given outliers. The results verify the superiority of DivOE on OOD detection over the direct adaptation of those strategies since they lack a critical diversification guidance. On the other hand, our framework has no strict constraints for the detailed manipulating operations. In other words, DivOE can also utilize those augmentation methods for extrapolation, for which we leave more discussion in Appendix C.

## 5 Conclusion

In this paper, we propose a novel learning framework, i.e., *Diversified Outlier Exposure* (DivOE), that promotes diversity in outlier exposure with the given auxiliary outliers. To empower the model with more knowledge about the decision boundary between ID and OOD data, DivOE conducts informative extrapolation guided by a newly designed objective. Through the diversification target, our method adaptively synthesizes meaningful outliers during the training process, which helps the model better shape the decision areas for OOD detection. We have conducted extensive experiments to demonstrate the effectiveness of our proposed DivOE and its compatibility with a range of OE-based methods, and also various ablation studies with further explorations to characterize the framework.

**Acknowledgments**

JNZ and BH were supported by the NSFC Young Scientists Fund No. 62006202, NSFC General Program No. 62376235, Guangdong Basic and Applied Basic Research Foundation No. 2022A1515011652, RIKEN Collaborative Research Fund, HKBU Faculty Niche Research Areas No. RC-FNRA-IG/22-23/SCI/04, and HKBU CSD Departmental Incentive Scheme. GY and JCY were supported by the National Key R&D Program of China (No. 2022ZD0160703), 111 plan (No. BP0719010) and National Natural Science Foundation of China (No. 62306178). TLL was partially supported by the following Australian Research Council projects: FT220100318, DP220102121, LP220100527, LP220200949, and IC190100031.

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

## Appendix for DivOE

The whole Appendix is organized as follows. In Appendix A, we present the detailed definitions and implementation of those post-hoc score functions and several OE-based methods that are considered in our exploration. In Appendix B, we formally present the theoretical analysis about DivOE. In Appendix C, we provide our extra experimental details and more comprehensive results with further discussion on the underlying implications. Finally, in Appendix D, we discuss the potential broader impact and limitations of our work.

## Reproducibility Statement

Below we summarize several important aspects to facilitate reproducible results:

- **Datasets.** The datasets we used are all publicly accessible, which is introduced in Section 4.1. For methods involving auxiliary outliers, we strictly follow previous works [Liu et al., 2020, Sun et al., 2021] to avoid overlap between the auxiliary dataset (Tiny-ImageNet) and any other OOD datasets.
- **Assumption.** We set our experiments to a fine-tuning scenario where a well-trained model on the original classification task is available, and the training samples are also available for subsequent fine-tuning [Hendrycks et al., 2019].
- **Open source.** We open our source code in the repository: DivOE
- **Environment.** All experiments are conducted with multiple runs on NVIDIA GeForce RTX 3090 GPUs with Python 3.7 and PyTorch 1.12.

## A    Details about Considered Baselines and Metrics

In this section, we provide the details about the baselines for the scoring functions and fine-tuning with auxiliary outliers, as well as the corresponding hyper-parameters and other related metrics that are considered in our work. Additionally, we provide further discussion on the comparison with two related works, e.g., MixOE and DOE, and also present the critical differences between the diversifying with previous concept of informative.

**Maximum Softmax Probability (MSP).**    [Hendrycks and Gimpel, 2017] proposes to use maximum softmax probability to discriminate ID and OOD samples. The score is defined as follows,

$$S_{\text{MSP}}(x; f) = \max_c P(y = c|x; f) = \max \text{softmax}(f(x)), \tag{12}$$

where $f$ represents the given well-trained model and $c$ is one of the classes $\mathcal{Y} = \{1, \ldots, C\}$. The larger softmax score indicates the larger probability for a sample to be ID data, reflecting the model's confidence on the sample.

**ODIN.**    [Liang et al., 2018] designed the ODIN score, leveraging the temperature scaling and tiny perturbations to widen the gap between the distributions of ID and OOD samples. The ODIN score is defined as follows,

$$S_{\text{ODIN}}(x; f) = \max_c P(y = c|\tilde{x}; f) = \max \text{softmax}(\frac{f(\tilde{x})}{T}), \tag{13}$$

where $\tilde{x}$ represents the perturbed samples (controled by $\epsilon$), $T$ represents the temperature. For fair comparison, we adopt the suggested hyperparameters [Liang et al., 2018]: $\epsilon = 1.4 \times 10^{-3}$, $T = 1.0 \times 10^4$.

**Mahalanobis.**    [Lee et al., 2018b] introduces a Mahalanobis distance-based confidence score, exploiting the feature space of the neural networks by inspecting the class conditional Gaussian distributions. The Mahalanobis distance score is defined as follows,

$$S_{\text{Mahalanobis}}(x; f) = \max_c -(f(x) - \hat{\mu}_c)^T \hat{\Sigma}^{-1}(f(x) - \hat{\mu}_c), \tag{14}$$

where $\hat{\mu}_c$ represents the estimated mean of multivariate Gaussian distribution of class $c$, $\hat{\Sigma}$ represents the estimated tied covariance of the $C$ class-conditional Gaussian distributions.

**Energy.** [Liu et al., 2020] proposes to use the Energy of the predicted logits to distinguish the ID and OOD samples. The Energy score is defined as follows,

$$S_{\text{Energy}}(x; f) = -T \log \sum_{c=1}^{C} e^{f(x)_c/T},$$ (15)

where $T$ represents the temperature parameter. As theoretically illustrated in Liu et al. [2020], a lower Energy score indicates a higher probability for a sample to be ID. Following [Liu et al., 2020], we fix the $T$ to 1.0 throughout all experiments.

**ASH.** [Djurisic et al., 2023] designs a extremely simple, post-hoc method called Activation SHaping for OOD detection. It removes a large portion of an input's activation at a late layer and adjusts the rest of the activation values by scaling them up or assigning them a constant value. The simplified representation is then passed throughout the rest of the network. The logit output is used to classify ID samples and calculate scores for OOD detection as usual. We adopt the energy score and apply the ASH-S version with the hyperparameter $p = 95$ for CIFAR-10 and $p = 85$ for CIFAR-100, as suggested by [Djurisic et al., 2023].

**Outlier Exposure (OE).** [Hendrycks et al., 2019] initiates a promising approach towards OOD detections by involving outliers to force apart the distributions of ID and OOD samples. In the experiments, we use the cross-entropy from $f(x_{\text{out}})$ to the uniform distribution as the $\mathcal{L}_{\text{OE}}$ [Lee et al., 2018a],

$$\mathcal{L}_f = \mathbb{E}_{\mathcal{D}_{\text{in}}} \left[ \ell_{\text{CE}}(f(x), y) \right] + \lambda \mathbb{E}_{\mathcal{D}_{\text{out}}^s} \left[ \log \sum_{c=1}^{C} e^{f(x)_c} - \mathbb{E}_{\mathcal{D}_{\text{out}}^s}(f(x)) \right].$$ (16)

**Energy (w. $\mathcal{D}_{\text{aux}}$).** In addition to using the Energy as a post-hoc score to distinguish ID and OOD samples, [Liu et al., 2020] proposes an Energy-bounded objective to further separate the two distributions. The OE objective is as follows,

$$\mathcal{L}_{\text{OE}} = \mathbb{E}_{\mathcal{D}_{\text{in}}^s}(\max(0, S_{\text{Energy}}(x, f) - m_{\text{in}}))^2 + \mathbb{E}_{\mathcal{D}_{\text{out}}^s}(\max(0, m_{\text{out}} - S_{\text{Energy}}(x, f)))^2.$$ (17)

We keep the thresholds same to [Liu et al., 2020]: $m_{\text{in}} = -23.0$ for CIFAR-10 and $m_{\text{in}} = -25.0$ for CIFAR-100, $m_{\text{out}} = -5.0$ for both CIFAR-10 and CIFAR-100.

**NTOM.** [Chen et al., 2021] greedily exploits informative auxiliary data to tighten the decision boundary between ID and OOD samples. It samples hard outliers with lower OOD scores to regularize the model for OOD detection. For a fair comparison, we adopt the energy score for both outlier sampling and model training and choose the suggested sampling parameter $p = 0$ [Chen et al., 2021].

**POEM.** [Ming et al., 2022] explores the Thompson sampling strategy [Thompson, 1933] to make the most use of outliers to learn a tight decision boundary. The details of Thompson sampling can refer to Ming et al. [2022].

**MixOE.** [Zhang et al., 2023] proposes to mix ID data and auxiliary outliers to cover the broad region where fine-grained OOD samples lie. The model is then trained such that the prediction confidence linearly declines as the input shifts from ID distribution to OOD distribution. Considering [Zhang et al., 2023] chooses different OOD test datasets from our work, we set the hyperparameters $\alpha = 1$ and $\beta = 0.1$ which are verified to be optimal by our tuning experiments.

**DOE.** [Wang et al., 2023] proposes to implicitly synthesize virtual outliers via model perturbation, which enhances the generalization of OE to better optimize the learning target [Foret et al., 2021].

### A.1 A Closer Look at Comparison with MixOE and DOE

In this part, we discuss more about the difference of our DOE with previous MixOE and DOE.

Conceptually, DivOE has a different underlying motivation from MixOE and DOE. Our DivOE is proposed for diversifying outlier exposure by extrapolating auxiliary outliers. In comparison, MixOE focuses on enhancing the fine-grained OOD detection by interpolation between ID samples and OOD

samples; DOE focuses on improving the generalization of the original outlier exposure by exploring model-level perturbation.

Technically, DivOE has a different algorithmic design from the others. Specifically, DivOE was proposed as a general framework for outlier extrapolation, which bases on OOD samples. It is different from MixOE which directly mix-up ID and OOD samples; It is also different from DOE that constructs the model perturbation for better optimization. It is worthy noting our DivOE is from a different perspective (data-level) compared with DOE (model-level), and they are orthogonal and combinable. DOE regards regularizing training on the same outliers with different weight perturbations as implicitly synthesized but does not change the outliers. The optimization of DOE is constrained by the original training samples and is also sensitive to tuning the model perturbation.

Table 3: Comparison of DivOE and DOE across different ID classification tasks.

| Dataset | Method | FPR95↓ | AUROC↑ | AUPR↑ | ID-ACC↑ |
|---------|--------|--------|--------|--------|---------|
| ImageNet | OE | 61.91 | 81.57 | 95.49 | **75.90** |
| | DivOE (OE backbone) | **60.01** | **81.96** | **95.59** | 75.74 |
| | DOE | 65.90 | 79.32 | 90.61 | **75.38** |
| | DivOE (DOE backbone) | **64.23** | **79.46** | **91.21** | 75.10 |
| SVHN | OE | 3.99 | 99.33 | 99.85 | **91.86** |
| | DivOE (OE backbone) | **2.24** | **99.60** | **99.91** | 91.72 |
| | DOE | 8.93 | 97.47 | 99.44 | 91.11 |
| | DivOE (DOE backbone) | **4.24** | **98.68** | **99.62** | 91.50 |
| CIFAR100 | OE | 27.67 | 91.89 | 97.93 | **75.41** |
| | DivOE (OE backbone) | **24.80** | **92.91** | **98.22** | 75.24 |
| | DOE | 30.69 | 93.13 | 98.39 | 71.41 |
| | DivOE (DOE backbone) | **20.26** | **95.41** | **98.91** | 71.43 |
| CIFAR10 | OE | 13.76 | 97.53 | 99.43 | 94.51 |
| | DivOE (OE backbone) | **11.66** | **97.82** | **99.48** | **94.66** |
| | DOE | 7.39 | 98.29 | 99.58 | 92.82 |
| | DivOE (DOE backbone) | **6.51** | **98.60** | **99.65** | 93.45 |

Empirically, as shown in Table 1, we have directly compared MixOE with our DivOE. The results show its performance is worse than our DivOE or even the original OE. For DOE, since it is from a different perspective and orthogonal, we compare DOE mainly in Table 2 of the original submission, and here we present Table 3 for further discussion. It shows: (1) DOE is sensitive to the ID classification task and even performs worse than the original OE in CIFAR-100/SVHN/ImageNet; (2) DivOE improves both OE and DOE under the corresponding backbones; (3) Even under unfair comparison between DivOE (OE backbone) vs. DOE, DivOE (OE backbone) still achieves the comparable or better performance on ImageNet/SVHN/CIFAR100 datasets.

## A.2 Conceptual Comparison of Diversified and Informative

In this part, we provide further discussion on the comparison of diversified and informative.

Conceptually, "diversified" is proposed to characterize one internal property of auxiliary outliers, while the "informative" in [Chen et al., 2021] is a concept for relative comparison. The intuition behind "diversified" is that we hope the outliers can span more informative areas beyond the original ones, unlike sampling the most "informative" outliers (i.e., near-boundary OOD samples that are close to ID data). Intuitively, Group 1 in Figure 2 indicates the auxiliary outliers which span more informative areas among those other groups. In other words, the most diversified Group 1 also covers the areas of other groups. Here we also utilize the Maximum Mean Discrepancy (MMD) [Borgwardt et al., 2006] to show the difference between "diversified" in our submission and "informative" in Table 4. We measure the discrepancy between the sampled/manipulated outliers with the original ones, termed as MMD[new, original], to indicate the "informative"; measure the discrepancy half by half within the sampled/manipulated outliers, termed as MMD[half new, half new], to indicate the "diversified". The results show that DivOE owns lower "informative" due to the partial outlier extrapolation but higher "diversified" on the inner dispersion of the outliers.

Technically, in our DivOE, "diversified" guides the training to expand the auxiliary outliers to the extrapolated distributions but also consider the original ones. We re-summarized the results

Table 4: "Diversified" and "Informative" measured by MMD scores.

| Dataset | Method | MMD[new, original] (informative) | MMD[half new, half new] (diversified) |
|---------|--------|----------------------------------|----------------------------------------|
| CIFAR100 | OE | - | $0.059 \pm 0.001$ |
| | NTOM* | $0.032 \pm 0.000$ | $0.056 \pm 0.000$ |
| | ATOM | $\mathbf{0.088} \pm 0.001$ | $0.056 \pm 0.000$ |
| | DivOE | $0.053 \pm 0.001$ | $\mathbf{0.113} \pm 0.002$ |
| CIFAR10 | OE | - | $0.052 \pm 0.000$ |
| | NTOM* | $0.030 \pm 0.000$ | $0.049 \pm 0.001$ |
| | ATOM | $\mathbf{0.285} \pm 0.005$ | $0.069 \pm 0.001$ |
| | DivOE | $0.118 \pm 0.001$ | $\mathbf{0.304} \pm 0.007$ |

of Figure 4(a) to Table 5 and show that our DivOE with the pursuit of "diversified" (conduct partial extrapolation) performs better than those totally considered "informative" (conduct total extrapolation). The results demonstrate the effectiveness and rationality of DivOE considering partial extrapolation for diversification.

Table 5: Comparison of DivOE pursuing "diversified" and "informative"

| Dataset | Method | FPR95↓ | AUROC↑ | AUPR↑ | ID-ACC↑ |
|---------|--------|--------|--------|-------|---------|
| CIFAR100 | OE | 27.67 | 91.89 | 97.93 | 75.41 |
| | DivOE (r=1.0 for "informative") | 43.18 | 88.11 | 97.11 | **75.43** |
| | DivOE (r=0.5 for "diversified") | **24.80** | **92.91** | **98.22** | 75.24 |
| CIFAR10 | OE | 13.76 | 97.53 | 99.43 | 94.51 |
| | DivOE (r=1.0 for "informative") | 22.05 | 96.24 | 99.14 | **94.81** |
| | DivOE (r=0.5 for "diversified") | **11.66** | **97.82** | **99.48** | 94.66 |

# B    Theoretical Analysis and Discussion

In this section, we formally present the theoretical implications with detailed definitions and assumptions. The complete theoretical analysis is based on the perspectives of sample complexity and OOD distribution gap in the previous works [Ming et al., 2022, Fang et al., 2022]. Different from the existing methods, we focus on the informative extrapolation proposed in our DivOE for diversifying the given auxiliary outliers towards the ID decision area. Intuitively, we provide an illustration of the conceptual comparison in Figure 5 to explain the differences. Different from the previous sampling-based methods which assume that the sampling space of the surrogate OOD distribution can be broad enough to cover the OOD decision area, our DivOE is targeted to conduct informative extrapolation to extend the given outlier exposure towards the ID decision area. In the following parts, we first introduce some preliminary setups and notations for the analyses.

**Preliminary setups and notations.**    Following the prior works [Lee et al., 2018a, Sehwag et al., 2021, Ming et al., 2022], we assume the extracted feature approximately follows a Gaussian mixture model (GMM) with the equal class priors as $\frac{1}{2}\mathcal{N}(\mu, \sigma^2\mathcal{I}) + \frac{1}{2}\mathcal{N}(-\mu, \sigma^2\mathcal{I})$. To be specific, $\mathcal{D}_{\text{in}} = \mathcal{N}(\mu, \sigma^2\mathcal{I})$ and $\mathcal{D}_{\text{out}}^s = \mathcal{N}(-\mu, \sigma^2\mathcal{I})$. Considering the hypothesis class as $\mathcal{H} = \text{sign}(\theta^T x), \theta \in \mathbb{R}^d$. The classifier outputs 1 if $x \sim \mathcal{D}_{\text{in}}$ and outputs -1 if $x \sim \mathcal{D}_{\text{out}}^s$.

First, we introduce the assumption about informative extrapolation on $\mathcal{D}_{\text{out}}^s$. The assumption is empirically supported to be valid by the previous exploration on the informative extrapolation in Figures 2 and 3, where the extrapolated outliers are more informative and close to the decision boundary towards ID samples.

**Assumption B.1** (Informative Extrapolation on $\mathcal{D}_{\text{out}}^s$)**.** Considering the extrapolated version $\mathcal{D}_{\text{out}}^e$ of the $\mathcal{D}_{\text{out}}^s$ in our proposed DivOE, we assume that the data points $x \sim \mathcal{D}_{\text{out}}^e$ satisfy the extended constraint based on the boundary scores $-|f_{\text{outlier}}(x)|$ defined in POEM Ming et al. [2022]:

$$\sum_{i=1}^{n} f_{\text{outlier}} \leq (\alpha - \tau(\epsilon))n, \tag{18}$$

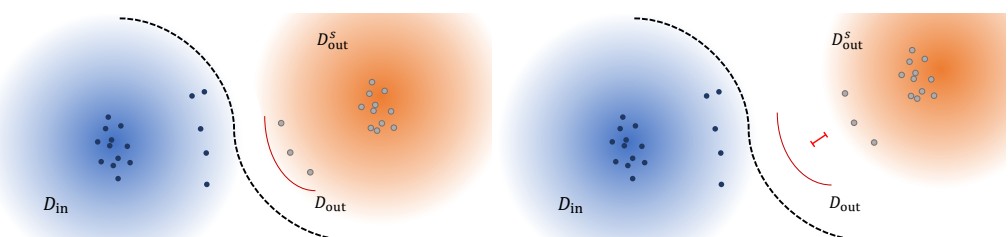

Figure 5: Illustration about the conceptual comparison of the previous assumption on auxiliary outliers in OE and that studied in our work. Left panel: the benign assumption on the auxiliary outliers, in which the surrogate OOD distribution $\mathcal{D}_{\text{out}}^s$ can be broad enough to cover the informative boundary outliers close to the decision area of ID data (in $\mathcal{D}_{\text{in}}$). Right panel: a more general assumption on the auxiliary outliers, considering that they can not well represent the unseen OOD distribution $\mathcal{D}_{\text{out}}$ since it is impractical or even infeasible to accurately pre-define and collect those boundary outliers.

where the $f_{\text{outlier}}$ is a function parameterized by some unknown ground truth weights and maps the high-dimensional input x into a scalar. Generally, it represents the discrepancy between the $\mathcal{D}_{\text{out}}^e$ and the true $\mathcal{D}_{\text{out}}$, indicated with the constraint $\alpha - \tau(\epsilon)$ results from the informative extrapolation. The $\tau(\epsilon)$ represents the extended diversity, which is positively related to the extrapolated strength $\epsilon$ in general.

Given the above assumption, we can derive the following extended lemma based on that adopted in POEM [Ming et al., 2022].

**Lemma B.2** (Constraint of Extrapolated $\mathcal{D}_{\text{out}}^e$). *Assume the data points $x \sim$ extrapolated $\mathcal{D}_{\text{out}}^e$ satisfy the following constraint for resulting in the following varied boundary margin: $\sum_{i=1}^n |2x_i^T \mu| \leq n\sigma^2(\alpha - \tau(\epsilon))$.*

*proof of Lemma. B.2.* Given the Gaussian mixture model described in the previous setup, we can obtain the following expression by Bayes' rule of $\mathbb{P}(\text{outlier}|x)$,

$$\mathbb{P}(\text{outlier}|x) = \frac{\mathbb{P}(x|\text{outlier})\mathbb{P}(\text{outlier})}{\mathbb{P}(x)} = \frac{1}{1 + e^{-\frac{1}{2\sigma^2}(d_{\text{outlier}}(x) - d_{\text{in}}(x))}}, \tag{19}$$

where $d_{\text{outlier}}(x)) = (x + \mu)^\top(x + \mu)$, $d_{\text{in}}(x) = (x - \mu)^\top(x - \mu)$, and $\mathbb{P}(\text{outlier}|x) = \frac{1}{1 + e^{-f_{\text{outlier}}(x)}}$ according to its definition. Then we have:

$$-f_{\text{outlier}} = -\frac{1}{2\sigma^2}(d_{\text{outlier}}(x) - d_{\text{in}}(x)), \tag{20}$$

$$-|f_{\text{outlier}}| = -\frac{1}{2\sigma^2}|(x - \mu)^\top(x - \mu) - (x + \mu)^\top(x + \mu)| = -\frac{2}{\sigma^2}|x^\top \mu|. \tag{21}$$

Therefore, we can get the constraint as: $\sum_{i=1}^n |2x_i^T \mu| \leq n\sigma^2(\alpha - \tau(\epsilon))$. $\qquad \square$

With the previous assumption and lemma, then we present the analysis as below.

**Complexity analysis on $\mathcal{D}_{\text{out}}^e$.** With the above lemma and the assumptions of informative extrapolation, we can derive the results to understand the benefit of our proposed DivOE.

Consider the given classifier defined as $\theta_{n_1,n_2}^* = \frac{1}{n_1+n_2}(\sum_{i=1}^{n_1} x_i^1 - \sum_{i=1}^{n_2} x_i^2)$, assume each $x_i^1$ is drawn *i.i.d.* from $\mathcal{D}_{\text{in}}$ and each $x_i^2$ is drawn *i.i.d* from $\mathcal{D}_{\text{out}}^e$, and assume the signal/noise ratio is $\frac{||\mu||}{\sigma} = r_0 \gg 1$, the dimentionality/sample size ratio is $\frac{d}{n} = r_1$, as well as exist some constant $\alpha < 1$. By decomposition, we can rewrite $\theta_{n_1,n_2}^* = \mu + \frac{n_1}{n_1+n_2}\theta_1 + \frac{n_2}{n_1+n_2}\theta_2$ with the following $\theta_1$ and $\theta_2$:

$$\theta_1 = \frac{1}{n_1}(\sum_{i=1}^{n_1} x_i^1) - \mu, \quad \theta_2 = \frac{1}{n_2}(-\sum_{i=1}^{n_2} x_i^2) - \mu, \tag{22}$$

Since $\theta_1 \sim \mathcal{N}(0, \frac{\sigma^2}{n_1}\mathcal{I})$, we have that $||\theta_1||^2 \sim \frac{\sigma^2}{n_1}\mathcal{X}_d^2$ and $\frac{\mu^T\theta_1}{||\mu||} \sim \mathcal{N}(0, \frac{\sigma^2}{n_1})$ to form the standard concentration bounds as:

$$\mathbb{P}(||\theta_1||^2 \geq \frac{\sigma^2}{n_1}(d + \frac{1}{\sigma})) \leq e^{-\frac{d}{8\sigma^2}}, \quad \mathbb{P}(\frac{|\mu^T\theta_1|}{||\mu||} \geq (\sigma||\mu||)^{\frac{1}{2}}) \leq 2e^{-\frac{n_1||\mu||}{2\sigma}} \tag{23}$$

The distribution of $\theta_2$ can be treated as a truncated distribution of $\theta_1$ as $x_i^2$ drawn *i.i.d.* from the extrapolated $\mathcal{D}_{\text{out}}^e$. Without losing the generality, we replace $n_1$ with $n$, and have the following inequality with a finite positive constant $a$:

$$\mathbb{P}(||\theta_2||^2 \geq \frac{\sigma^2}{n_1}(d + \frac{1}{\sigma})) \leq ae^{-\frac{d}{8\sigma^2}} \tag{24}$$

According to Lemma B.2, we can have that $|\mu^T\theta_2| \leq ||\mu||^2 + \frac{\sigma^2(\alpha - \tau(\epsilon))}{2}$. Now we can have $||\theta_1||^2 \leq \frac{\sigma^2}{n}(d + \frac{1}{\sigma}), ||\theta_2||^2 \leq \frac{\sigma^2}{n}(d + \frac{1}{\sigma}), \frac{|\mu^T\theta_1|}{||\mu||} \leq (\sigma||\mu||)^{\frac{1}{2}}$ simultaneously hold and derive the following recall the decomposition,

$$||\theta_{n_1,n_2}^*||^2 = ||\mu + \frac{n_1}{n_1+n_2}\theta_1 + \frac{n_2}{n_1+n_2}\theta_2||^2 \leq \frac{\sigma^2}{n}(d + \frac{1}{\sigma}) + ||\mu||^2, \tag{25}$$

and

$$|\mu^T\theta_{n_1,n_2}^*| \geq \frac{1}{2}(||\mu||^2 - \sigma^{\frac{1}{2}}||\mu||^{\frac{3}{2}} - \frac{\sigma^2(\alpha - \tau(\epsilon))}{2}). \tag{26}$$

With the above inequality derived in Eq. (25) and Eq. (26), we can have the following bound with the probability at least $1 - (1 + a)e^{-\frac{r_1 n}{8\sigma^2}} - 2e^{-\frac{n_1||\mu||}{2\sigma}}$

$$\frac{\mu^T\theta_{n_1,n_2}^*}{\sigma||\theta_{n_1,n_2}^*||} \geq \frac{||\mu||^2 - \sigma^{\frac{1}{2}}||\mu||^{\frac{3}{2}} - \frac{\sigma^2(\alpha - \tau(\epsilon))}{2}}{2\sqrt{\frac{\sigma^2}{n}(d + \frac{1}{\sigma}) + ||\mu||^2}} \tag{27}$$

Since $\text{FPR}(\theta_{n_1,n_2}^*) = erf(\frac{\mu^T\theta_{n_1,n_2}^*}{\sigma||\theta_{n_1,n_2}^*||})$ is monotonically decreasing, as the lower bound of $\frac{\mu^T\theta_{n_1,n_2}^*}{\sigma||\theta_{n_1,n_2}^*||}$ will increase as the constraint from the extrapolated $\mathcal{D}_{\text{out}}^e$ changed accordingly in our DivOE, the upper bound of $\text{FPR}(\theta_{n_1,n_2}^*)$ will decrease. From the above analysis, we can observe the benefit of DivOE when conducting the informative extrapolation.

**Additional Discussion with Extrapolation Condition.** The extrapolation of outlier data towards inliers may not always be possible. Similar to the condition analysis in [Fang et al., 2022], the informative extrapolation that targets better characterize the decision boundary between ID and OOD also needs some requirements or realization assumptions. Intuitively, it mainly relies on auxiliary outliers and specific extrapolation techniques. For the former, the original outliers should not be far away from the ID sample; For the latter, the extrapolation techniques should be able to add enough perturbation or semantic changes to the original outlier. It is worthy further exploring the condition in a theoretical way by referring the analysis in Fang et al. [2022].

## C  Additional Experimental Results and Further Discussion

In this section, we provide more experiment results from different perspectives to characterize our proposed DivOE. First, we introduce the additional experimental setups for the empirical verification in previous figures and our learning framework. Second, we provide more analyses of the manipulated data via informative extrapolation. Finally, comprehensive results with discussions are provided.

### C.1  Additional Experimental Setups

**Figure 1.** In the right-panel of Figure 1, we conduct experiments to evaluate the OOD detection performance of different OE-based methods with the gradually reduced number of auxiliary outliers. We first sample a particular number of the least informative auxiliary outliers to simulate the different cases in collecting those surrogate OOD data. With the highest OOD scores, we sample the selected auxiliary outliers from the whole auxiliary dataset. Specifically, we choose the negative boundary

score generated by POEM [Ming et al., 2022] as the OOD score. For the fair comparison, we adjust the size of the mini-batch for OOD samples so that all methods actually train on 90% of the least informative auxiliary dataset by random sampling, or their custom sampling strategies during every epoch while keeping the number of ID training data unchanged.

**Figure 2.** In the left-middle panel of Figure 2, we evaluate the OOD detection performance of Energy-OE [Liu et al., 2020] with auxiliary outliers with different level of diversity. To simulate the collected auxiliary outliers from different levels of diversity, we first sample the 40000 least informative outliers from Tiny-ImageNet [Le and Yang, 2015] according to the boundary score generated by POEM [Ming et al., 2022]. Then we choose 10000 informative outliers in different but not-overlapped boundary score intervals, namely $[0, 10000], [10000, 20000], [30000, 40000], [40000, 50000], [50000, 60000]$, and combine them with the 40000 least informative outliers to form 5 groups of auxiliary outliers with different levels of diversity. In the two right panels of Figure 2, we present TSNE visualization of feature embedding (taken at the penultimate layer of the network) on ID, the original outliers, and extrapolated outliers after the first training epoch in DivOE, with extrapolated ratio $r = 0.5$, extrapolate epsilon $\epsilon = 0.05$ and optimization step number $k = 5$.

**Figure 3.** In the two left panel of Figure 3, we keep the extrapolated ratio $r$ fixed to 0.5 and optimization step number $k$ fixed to 5. In the right-middle panel, we compare ID/OOD distributions based on Energy score after training with DivOE, with hyperparameters $r = 0.5, \epsilon = 0.05$, and $k = 5$.

**Figure 4.** In Figure 4(a), the extrapolated epsilon $\epsilon$ is set to 0.5 and optimization step number $k$ is set to 5. In Figure 4(b), we fix the extrapolated ratio $r$ to 0.5 and optimization step number $k$ to 5.

**Training details.** We conduct all major experiments on pre-trained WideResNet [Zagoruyko and Komodakis, 2016]with 40 depth and 2 widen factor and fix the number of fine-tuning epochs to 10, following the previous research work [Hendrycks et al., 2019, Liu et al., 2020]. The models are trained using stochastic gradient descent [Kiefer and Wolfowitz, 1952] with Nesterov momentum [Duchi et al., 2011]. We adopt Cosine Annealing [Loshchilov and Hutter, 2017] to schedule the learning rate, which begins at $0.001$. We set the momentum and weight decay to be $0.9$ and $10^{-4}$ respectively for all experiments. The size of the mini-batch is $128$ for both ID samples and OOD samples when training and $200$ for both when testing.

**Hyper-parameter of DivOE.** Throughout the paper, all the experiments with DivOE are conducted with extrapolated ratio $r = 0.5$, extrapolate epsilon $\epsilon = 0.05$, and optimization step number $k = 5$, unless stated otherwise.

### C.2 Analyses on the Extrapolated Data

In this part, we further compare the extrapolated outliers with the given auxiliary outliers during the fine-tuning process of OE. In Figure 6, we adopt the TSNE visualization to check the feature representation of the extrapolated outliers during training. Through the comparison with the given outliers (represented by the blue circles), the extrapolated outliers (represented by the orange cricles) manipulated by our introduced target (e.g., Eq. (6)) can better cover the decision area between ID and OOD data. Note that the extrapolated OOD samples are optimized to be closer to the ID samples (represented by the green circles), which indicates this extrapolated portion can perform better on more accurately shaping the decision boundary of the binary classification task for ID and OOD.

In Figure 8, we present the energy score of the extrapolated outliers compared with the original ones. It can be found that the newly synthesized outliers can generally easier to be recognized as ID samples than the original auxiliary outliers. In Figure 9 and Table 28, we report the loss differences corresponding to the maximization part that verifies our operation indeed achieves the conceptual idea of diversified outlier exposure. Intuitively, we visualize the extrapolated outliers in Figure 10. The extrapolated samples still keep the semantic information that belongs to the OOD class space.

In addition, we also conduct another quantitative investigation on the extrapolated outliers with the original ones. In Table 6, we adopt the Maximum Mean Discrepancy (MMD) [Borgwardt et al., 2006] to check the distance between the distributions of the given auxiliary outliers with the extrapolated ones. By varying the specific $\epsilon$ that is adopted in the multi-step optimization (e.g., Eq. (9)), we can

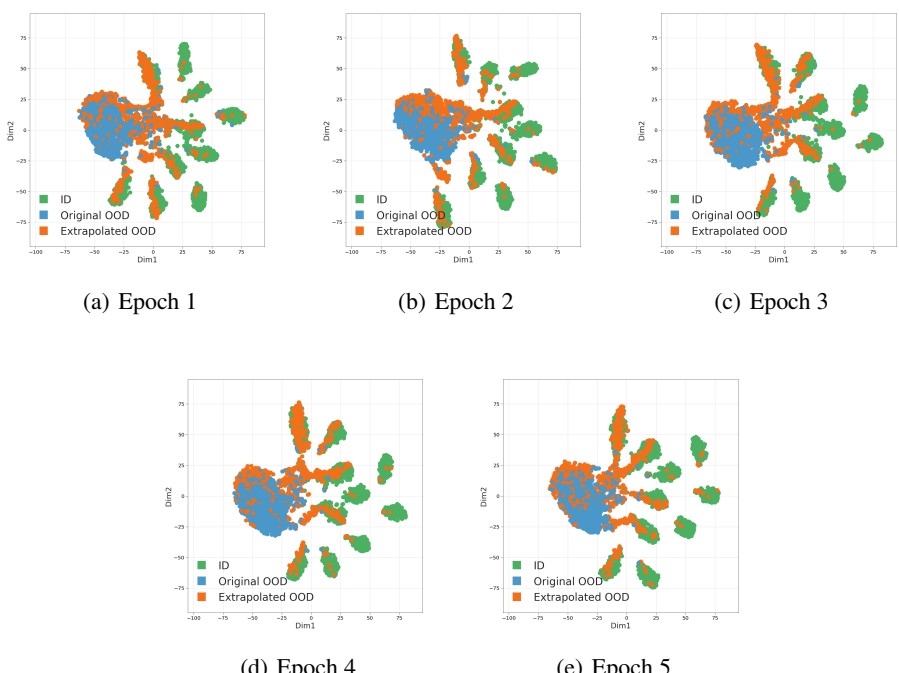

|                |                |                |
| :------------: | :------------: | :------------: |
| (a) Epoch 1    | (b) Epoch 2    | (c) Epoch 3    |
|                | (d) Epoch 4    | (e) Epoch 5    |

Figure 6: TSNE visualization of CIFAR10 ID data and the extrapolated outliers generated during OE.

observe that the manipulated outliers with large $\epsilon$ perform large discrepancy with the original outliers. It is intuitive and reasonable with more changes on semantic levels as shown in the left panel of Figure 3. However, as previously illustrated in the left-middle panel of Figure 3, we also find the larger $\epsilon$ may not always indicates more informative outliers as the semantic information may be lost when the pixel perturbations are arbitrarily large. It is worth to be further explored in the future.

Table 6: MMD value between original outliers and the extrapolated outliers with different $\epsilon$ in DivOE

| $\epsilon$ | 0.01 | 0.025 | 0.05 | 0.075 | 0.1 | 0.15 | 0.5 |
| :--------: | :----: | :----: | :----: | :----: | :----: | :----: | :----: |
| MMD value | 0.0155 | 0.0176 | 0.0208 | 0.0233 | 0.0253 | 0.0286 | 0.0439 |

### C.3 Discussion on Using PGD-based Attack

Note that, PGD is a replaceable choice in the framework of DivOE. To better understand the effect of these attack-based noise, we compare it with other inner-attack methods and show that PGD with the uniform distribution loss is generally the better way to achieve higher performance.

We would like to explain that DivOE was proposed as a general framework for diversifying outlier exposure with partially conduct informative extrapolation, which allows any appropriate noise or augmentation techniques to be incorporated. To show the effectiveness of PGD, we conduct the following experiments. On the one hand, we compare the random noise, re-adversarial perturbation, and adversarial perturbation to justify its rationality. The results are summarized in the Table 7 and show the adversarial perturbation achieve better performance than other types of perturbations. On the other hand, we compare with other adversarial attacks and data augmentation methods for inner-maximization part in Table 8 and show that PGD with the uniform distribution loss is generally a better way to achieve higher performance due to the optimization originality.

**How would $\alpha$ play an effect in DivOE** As for $\alpha$ , it is a step size adopted in the informative extrapolation. To better understand its effect, we conducted the experiments by changing the step size in the PGD attack in Table 9. Since the step size is only related to the optimization of the constrained

Table 7: Rationality of adopting PGD attack in the inner maximization.

| Dataset | Method | Perturbation Direction & method | FPR95 | AUROC | AUPR | ID-ACC |
|---|---|---|---|---|---|---|
| CIFAR100 | OE | | 27.67 | 91.89 | 97.93 | 75.41 |
| | DivOE | Re-PGD (most uninformative) | 27.36 | 92.02 | 97.98 | 75.62 |
| | DivOE | Random Noise | 26.16 | 92.39 | 98.06 | 75.56 |
| | DivOE | Mixup | 25.70 | 92.46 | 98.07 | **76.06** |
| | DivOE | PGD (most informative) | **24.80** | **92.91** | **98.22** | 75.24 |
| CIFAR10 | OE | | 13.76 | 97.53 | 99.43 | 94.51 |
| | DivOE | Re-PGD (most uninformative) | 13.20 | 97.59 | 99.43 | 94.80 |
| | DivOE | Random Noise | 13.19 | 97.60 | 99.43 | 94.81 |
| | DivOE | Mixup | 12.87 | 97.59 | 99.42 | **94.83** |
| | DivOE | PGD (most informative) | **11.66** | **97.82** | **99.48** | 94.66 |

Table 8: DivOE incorporating different attacked noise

| Dataset | Method | Attack Method | FPR95↓ | AUROC↑ | AUPR↑ | ID-ACC↑ |
|---|---|---|---|---|---|---|
| CIFAR100 | OE | - | 27.67 | 91.89 | 97.93 | 75.41 |
| | DivOE | Random | 26.16 | 92.39 | 98.06 | 75.56 |
| | DivOE | FGSM | 25.58 | 92.60 | 98.14 | 75.59 |
| | DivOE | PGD | 24.80 | 92.91 | 98.22 | 75.24 |
| | DivOE | C&W | **24.20** | **93.02** | **98.25** | 75.18 |
| | DivOE | TRADES | 24.88 | 92.85 | 98.20 | 75.13 |
| | DivOE | Mixup | 25.70 | 92.46 | 98.07 | **76.06** |
| | DivOE | CutMix | 28.46 | 91.70 | 97.89 | 75.62 |
| | DivOE | AugMix | 25.40 | 92.60 | 98.12 | 75.58 |
| CIFAR10 | OE | - | 13.76 | 97.53 | 99.43 | 94.51 |
| | DivOE | Random | 13.19 | 97.60 | 99.43 | 94.81 |
| | DivOE | FGSM | 12.71 | 97.70 | 99.45 | 94.78 |
| | DivOE | PGD | **11.66** | **97.82** | 99.48 | 94.66 |
| | DivOE | C&W | 11.73 | 97.74 | 99.46 | 94.64 |
| | DivOE | TRADES | 12.47 | 97.67 | 99.44 | 94.54 |
| | DivOE | Mixup | 12.87 | 97.59 | 99.42 | **94.83** |
| | DivOE | CutMix | 13.95 | 97.54 | 99.42 | 94.78 |
| | DivOE | AugMix | 12.41 | **97.82** | **99.49** | 94.71 |

maximization, the effects on the final performance of DivOE are not significant. By enlarging the step size, we may save more extra time on the inner maximization to reduce the time cost relatively, which is beneficial for algorithm deployment.

Table 9: The effect of $\alpha$ in training of DivOE

| Dataset | Method | Attack Method | Attack Steps | FPR95↓ | AUROC↑ | AUPR↑ | ID-ACC↑ |
|---|---|---|---|---|---|---|---|
| CIFAR-100 | OE | - | - | 27.67 | 91.89 | 97.93 | 75.41 |
| | DivOE | PGD (alpha=0.005) | 20 | 25.02 | 92.69 | 98.15 | 75.27 |
| | DivOE | PGD (alpha=0.01) | 10 | 25.08 | 92.75 | 98.17 | 74.82 |
| | DivOE | PGD (alpha=0.02) | 5 | **24.41** | **92.99** | **98.23** | 75.34 |
| | DivOE | PGD (alpha=0.04) | **3** | 25.26 | 92.71 | 98.17 | **75.44** |
| CIFAR-10 | OE | - | - | 13.76 | 97.53 | 99.43 | 94.51 |
| | DivOE | PGD (alpha=0.005) | 20 | **11.92** | 97.78 | **99.47** | 94.57 |
| | DivOE | PGD (alpha=0.01) | 10 | 12.20 | 97.61 | 99.42 | 94.59 |
| | DivOE | PGD (alpha=0.02) | 5 | 12.11 | **97.80** | **99.47** | 94.64 |
| | DivOE | PGD (alpha=0.04) | **3** | 12.23 | 97.76 | 99.46 | **94.79** |

**Visual illustration of Extrapolation Process.** From the algorithmic perspective, the adversarial perturbation with uniform distribution loss is a non-target perturbation. The intuition is to perturb the OOD sample to be more like ID regarded by the model. Maximizing uniform distribution loss has no requirement for each OOD sample to be perturbed into some specific ID classes. Thus, it is actually automatically decided by the gradient for the specific perturbation directions. To better

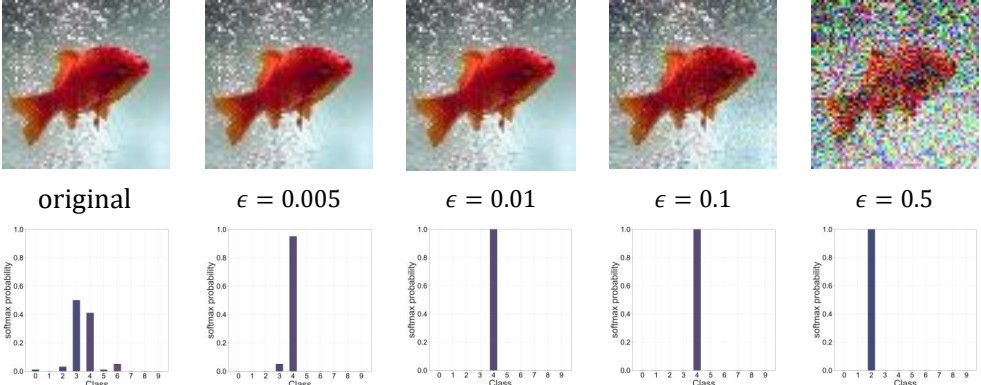

Figure 7: Prediction on the perturbed outlier with different extrapolation strength $\epsilon$.

understand the perturbation effect, we conduct the case study on the specific example in Figure 7 for their specific prediction results during the adversarial perturbation process. It can be find that after the adversarial perturbation, the prediction result of the second example will be concentrate on one specific class with high confidence. By enlarging the perturbation strength , the example will be perturbed to be confidently predicted on different classes.

## C.4 Rationality of Partially Extrapolation

Considering the initial motivation to diversify outliers, we explore to extrapolate more potential OOD distribution beyond the original auxiliary ones. For the unperturbed OOD samples used in training, we hope the training can still learn the knowledge from the original auxiliary outliers, for the perturbed OOD samples, we hope the training can find different but informative outliers beyond the original ones to improve the generalization of outlier exposure.

Formally, Eq. (5) is an instance-level realization of Eq. (4) that actually contains the perturbed part and the unperturbed part. The deduction from Eq. (4) to Eq. (5) requires to merge the same terms in Eq. (4) first, and then specifically set with a subsequent rescaling. Finally, we have the formulation that a portion of data to conduct maximization part for diversifying the existing data distribution and the remaining other portion of data corresponds to learning on the original outliers.

To better understand the specific effects of different samples, it can be referred to the left panel of Figure 4, in which we show the OOD detection performance of DivOE under different portions of perturbed data during training. Generally, using a small part of perturbed samples can achieve the extrapolation target and improve the performance. However, if we totally perturb all the data, it is even worse than the original OE.

Intuitively, we hope the extrapolation can realize the diversification target and cover more potential OOD distributions beyond the original auxiliary ones. Under such high-level guidance, generally, the extrapolation ratio is expected to be not too large to totally change the whole original auxiliary distribution, which is empirically demonstrated to be negative for OE. As for diversified strength, we kindly refer to the left middle panel of Figure 3, which serves as a search space for synthesizing different outliers using different. In practical hyper-parameter tuning, we may generally suggest starting with the original OE baseline (i.e., extrapolation ratio equals 0 and diversified strength equals 0) and enlarging the extrapolation ratio or diversified strength step by step, finding a stage where the OOD detection performance on the validation set will not be further enhanced. For the coarse-grained augmentations and specific OOD scores, it may be further decided by other.

For four extrapolation factors, we re-summarize the previous results and discuss the general guidance for tuning. First, MSP score and PGD are empirically demonstrated to be the appropriate choice for extrapolation, which can be fixed (refer to the results in Figure 4(c)). Second, extrapolation ratio and strength (i.e., $\epsilon$) have general trends either individually or jointly considering them in Table 10. The performance of the extrapolation ratio is monotonous regarding different $\epsilon$. The large values would degrade performance due to the loss of diversity on the auxiliary outliers. Then, the extrapolation strength could be tuned after the other factors are set to appropriate choices. Our empirical results

show that very small or large $\epsilon$ should be avoided, and epsilons from 0.05 to 0.1 can generally achieve satisfactory results by referring previous experimental results.

Table 10: DivOE with different extrapolation ratiosand strengths.

| Dataset | Method | Extrapolation ratio $r$ | Extrapolation strength $\epsilon$ | FPR95$\downarrow$ |
|---------|--------|------------------------|-----------------------------------|-------------------|
| CIFAR-100 | OE | - | - | 27.67 |
| | DivOE | 1 | 0.01 | 30.52 |
| | DivOE | 0.5 | 0.01 | 29.06 |
| | DivOE | 0.25 | 0.01 | 26.21 |
| | DivOE | 0.1 | 0.01 | 24.57 |
| | DivOE | 1 | 0.05 | 43.21 |
| | DivOE | 0.5 | 0.05 | 24.80 |
| | DivOE | 0.25 | 0.05 | 24.21 |
| | DivOE | 0.1 | 0.05 | 23.54 |
| | DivOE | 1 | 0.125 | 49.69 |
| | DivOE | 0.5 | 0.125 | 24.94 |
| | DivOE | 0.25 | 0.125 | 25.36 |
| | DivOE | 0.1 | 0.125 | 24.77 |

## C.5 Extra Experimental Results and Discussions

In this part, we present the extra experimental results in the following perspectives to characterize the learning framework of DivOE for the OOD detection task.

**Experiments on Add-on Modules.** For adding other methods to Table 2, we conduct experiments accordingly and summarize the results in Table 11. The results show that DivOE can consistently improve them on the original basis compared with other counterparts across different setups.

Table 11: Extra comparison with various add-on methods.

| Dataset | Method | FPR95$\downarrow$ | AUROC$\uparrow$ | AUPR$\uparrow$ | ID-ACC$\uparrow$ |
|---------|--------|-------------------|------------------|-----------------|-------------------|
| CIFAR-100 | OE | 27.67 | 91.89 | 97.93 | 75.41 |
| | DOE | 30.69 | 93.13 | 98.39 | 71.43 |
| | MixOE | 35.26 | 90.31 | 97.58 | 75.74 |
| | DivOE | 24.80 | 92.91 | 98.22 | 75.24 |
| | Energy | 27.86 | 90.70 | 97.40 | 75.41 |
| | Energy+DOE | 44.91 | 88.80 | 97.22 | 73.79 |
| | Energy+MixOE | 38.14 | 85.12 | 95.40 | 73.87 |
| | Energy+DivOE | 24.71 | 92.04 | 97.82 | 75.25 |
| CIFAR-10 | OE | 13.76 | 97.53 | 99.43 | 94.51 |
| | DOE | 7.39 | 98.29 | 99.58 | 92.82 |
| | MixOE | 13.57 | 97.47 | 99.39 | 94.66 |
| | DivOE | 11.66 | 97.82 | 99.48 | 94.66 |
| | Energy | 18.15 | 90.90 | 96.34 | 94.02 |
| | Energy+DOE | 9.65 | 97.70 | 99.37 | 94.83 |
| | Energy+MixOE | 31.90 | 83.72 | 93.50 | 93.72 |
| | Energy+DivOE | 16.03 | 92.23 | 96.87 | 94.53 |

**Experiments on Fine-grained OOD Test Sets.** In order to further understand the effectiveness of our DivOE on different OOD datasets, we report the fine-grained results of our experiments on CIFAR-10/CIFAR-100 with 5 OOD datasets (Textures, Places365, iSUN, LSUN-r, LSUN-c). The results on the 5 OOD datasets demonstrate the general effectiveness of our DivOE.

In Tables 12 and 13, we summarize the fine-grained detection performance. The results reflect the performance improvement of each method compared with those representative baselines. Our DivOE can significantly reduce the FPR95 score on some OOD testing sets in which those previous baselines do not perform well (e.g., Textures and Places365), and achieve comparable performance on the rest ones. Note that it is general to have some tradeoffs [Hendrycks et al., 2019, Chen et al., 2021, Ming et al., 2022, Fang et al., 2022] on the performance change when using different methods under the OE framework. DivOE shows a good balance in gaining significant improvement without sacrificing the other performance too much via the informative extrapolation. On the other hand, the significant

Table 12: OOD detection performance on CIFAR-10 [Krizhevsky, 2009] as ID. All methods are trained on the same backbone. Values are percentages. ↑ indicates larger values are better, and ↓ indicates smaller values are better. (averaged by multiple trials)

| Method | OOD dataset | | | | | | | | | | ID ACC |
| | Textures | | Places365 | | iSUN | | LSUN-r | | LSUN-c | | |
| | FPR95↓ | AUROC↑ | FPR95↓ | AUROC↑ | FPR95↓ | AUROC↑ | FPR95↓ | AUROC↑ | FPR95↓ | AUROC↑ | |
|---|---|---|---|---|---|---|---|---|---|---|---|
| MSP | 59.27 | 88.50 | 60.77 | 88.13 | 56.79 | 89.53 | 53.60 | 91.02 | 30.55 | 95.68 | 94.84 |
| ODIN | 54.84 | 80.28 | 48.89 | 85.66 | 28.68 | 93.85 | 23.17 | 95.05 | 9.96 | 97.90 | 94.84 |
| Mahalanobis | 16.99 | 96.69 | 69.96 | 82.54 | 31.62 | 94.44 | 32.41 | 94.70 | 30.50 | 95.06 | 94.84 |
| Energy | 53.17 | 85.40 | 40.22 | 89.91 | 34.60 | 92.44 | 28.48 | 93.85 | 8.12 | 98.35 | 94.84 |
| GradNorm | 73.17 | 58.11 | 78.27 | 60.46 | 70.28 | 70.82 | 65.63 | 73.23 | 11.41 | 97.02 | 94.84 |
| OE | 21.81 | 96.15 | 44.86 | 91.77 | 0.00 | 100.00 | 0.00 | 100.00 | 0.99 | 99.85 | 94.56 |
| Energy (w. $\mathcal{D}_{aux}$) | 33.59 | 85.47 | 58.39 | 68.32 | 0.00 | 100.00 | 0.00 | 100.00 | 0.31 | 99.83 | 94.50 |
| VOS | 55.45 | 81.72 | 46.89 | 87.02 | 4.21 | 99.13 | 33.33 | 92.47 | 39.16 | 90.82 | 95.02 |
| NTOM | 32.78 | 86.30 | 59.03 | 67.50 | 0.00 | 100.00 | 0.00 | 100.00 | 1.17 | 99.38 | 95.06 |
| POEM | 34.73 | 85.28 | 57.82 | 67.93 | 0.00 | 100.00 | 0.00 | 100.00 | 0.43 | 99.74 | 95.10 |
| MixOE | 22.23 | 95.74 | 39.45 | 92.86 | 0.26 | 99.92 | 0.27 | 99.92 | 4.40 | 99.21 | 95.00 |
| **DivOE** (Ours) | 12.28 | 97.73 | 44.97 | 91.51 | 0.07 | 99.98 | 0.02 | 99.99 | 3.03 | 99.47 | 94.56 |

Table 13: OOD detection performance on CIFAR-100 [Krizhevsky, 2009] as ID. All methods are trained on the same backbone. Values are percentages. ↑ indicates larger values are better, and ↓ indicates smaller values are better. (averaged by multiple trials)

| Method | OOD dataset | | | | | | | | | | ID ACC |
| | Textures | | Places365 | | iSUN | | LSUN-r | | LSUN-c | | |
| | FPR95↓ | AUROC↑ | FPR95↓ | AUROC↑ | FPR95↓ | AUROC↑ | FPR95↓ | AUROC↑ | FPR95↓ | AUROC↑ | |
|---|---|---|---|---|---|---|---|---|---|---|---|
| MSP | 82.90 | 73.53 | 82.31 | 74.08 | 83.28 | 75.10 | 83.06 | 74.08 | 66.41 | 83.85 | 75.96 |
| ODIN | 76.50 | 74.83 | 81.17 | 74.20 | 60.64 | 85.08 | 62.15 | 84.38 | 36.20 | 93.31 | 75.96 |
| Mahalanobis | 38.99 | 90.64 | 88.58 | 67.92 | 25.33 | 94.57 | 20.89 | 96.05 | 91.26 | 69.73 | 75.96 |
| Energy | 79.70 | 76.34 | 80.48 | 75.78 | 82.44 | 77.87 | 81.01 | 77.67 | 35.69 | 93.43 | 75.96 |
| GradNorm | 88.31 | 60.52 | 97.22 | 53.15 | 99.20 | 43.52 | 99.26 | 38.83 | 41.12 | 91.96 | 75.96 |
| OE | 54.11 | 84.69 | 77.42 | 76.27 | 0.07 | 99.99 | 0.04 | 100.00 | 4.58 | 99.06 | 75.50 |
| Energy (w. $\mathcal{D}_{aux}$) | 56.93 | 81.22 | 80.53 | 72.76 | 0.00 | 100.00 | 0.00 | 100.00 | 0.52 | 99.88 | 75.14 |
| VOS | 78.50 | 74.02 | 82.21 | 73.40 | 85.24 | 75.86 | 83.59 | 75.58 | 25.65 | 94.88 | 74.94 |
| NTOM | 59.65 | 81.76 | 80.40 | 72.46 | 0.00 | 100.00 | 0.00 | 100.00 | 2.24 | 99.46 | 75.31 |
| POEM | 51.72 | 84.16 | 79.94 | 72.84 | 0.00 | 100.00 | 0.00 | 100.00 | 0.34 | 99.92 | 75.88 |
| MixOE | 61.78 | 82.13 | 72.95 | 78.20 | 2.96 | 99.32 | 1.05 | 99.75 | 33.56 | 92.88 | 75.77 |
| **DivOE** (Ours) | 34.16 | 90.61 | 74.42 | 77.53 | 0.18 | 99.94 | 0.03 | 99.99 | 12.97 | 97.10 | 75.35 |

performance improvement of DivOE on the Textures and Places365 (that OOD test sets that the original OE does not perform well) exactly verifies that the informatively extrapolated outliers better represent the unseen OOD inputs that are close to the ID decision area.

**Experiments on Sample Comparison** It may be hard to align the sample number as different methods have different synthesis motivations and but we can actually present the number of generated samples. To facilitate the comparison, we summarize the specific outlier numbers that were manipulated in the adopted algorithm with its performance in OOD detection in Table 14. The results show that, compared with previous methods, either need go through all the outliers for sampling or change the training on all the auxiliary outliers, our DivOE can conduct extrapolation on a small portion of original outliers but achieve generally better performance across different ID classification tasks.

Table 14: Numbers of outliers manipulated or sampled during training

| Dataset | Method | Numbers | FPR95↓ | AUROC↑ | AUPR↑ | ID-ACC↑ |
|---|---|---|---|---|---|---|
| | OE | 0 | 27.67 | 91.89 | 97.93 | 75.41 |
| | NTOM | 50000 | 27.58 | 91.08 | 97.52 | 75.41 |
| CIFAR100 | POEM | 50000 | 26.50 | 91.33 | 97.58 | **75.74** |
| | MixOE | 50000 | 35.26 | 90.31 | 97.58 | **75.74** |
| | DivOE | **25000** | **24.80** | **92.91** | **98.22** | 75.24 |
| | OE | 0 | 13.76 | 97.53 | 99.43 | 94.51 |
| | NTOM | 50000 | 17.76 | 91.08 | 96.40 | **95.04** |
| CIFAR100 | POEM | 50000 | 17.81 | 90.96 | 96.36 | 94.90 |
| | MixOE | 50000 | 13.57 | 97.47 | 99.39 | 94.66 |
| | DivOE | **25000** | **11.66** | **97.82** | **99.48** | 94.66 |

**Experiments on Smaller Number of Auxiliary Samples.** Here, we added the experiments under different numbers of auxiliary outliers used for all OE-based methods. The experimental results are summarized in Table 15, which indicates that the performance of OOD detection can be generally better if we use more outliers. Under smaller number of samples, i.e., from 50000 to 1000 (relatively smaller than 50000 ID data), DivOE can consistently perform better than OE.

Table 15: Performance comparison by changing the numbers of auxiliary outliers.

| Dataset | Method | Number (ratio to ID samples) | FPR95 | AUROC | AUPR | ID-ACC |
|---|---|---|---|---|---|---|
| CIFAR100 | OE | 50000 (1.0) | 26.72 | 91.12 | 97.51 | 75.58 |
| | NTOM | | 29.18 | 90.72 | 97.45 | **75.77** |
| | POEM | | 27.20 | 90.79 | 97.41 | **75.77** |
| | MixOE | | 35.28 | 90.29 | 97.57 | 75.67 |
| | DivOE | | **26.27** | **92.71** | **98.18** | 75.34 |
| | OE | 5000 (0.1) | 33.64 | 89.98 | 97.36 | 75.92 |
| | NTOM | | 33.23 | 88.92 | 96.86 | 75.87 |
| | POEM | | 33.27 | 88.59 | 96.73 | 75.64 |
| | MixOE | | 37.32 | 89.39 | 97.31 | **76.09** |
| | DivOE | | **32.80** | **90.89** | **97.71** | 75.62 |
| | OE | 1000 (0.02) | 38.40 | 88.93 | 97.17 | **76.14** |
| | NTOM | | 38.11 | 89.12 | 97.45 | 76.12 |
| | POEM | | 37.69 | 89.47 | **97.56** | 75.98 |
| | MixOE | | 40.78 | 87.98 | 96.89 | 76.10 |
| | DivOE | | **36.74** | **89.63** | 97.36 | 76.08 |
| CIFAR10 | OE | 50000 (1.0) | 12.64 | **97.76** | **99.47** | 94.85 |
| | NTOM | | 19.84 | 90.12 | 96.23 | 94.80 |
| | POEM | | 20.21 | 90.15 | 96.10 | **94.97** |
| | MixOE | | 14.26 | 97.31 | 99.35 | 94.92 |
| | DivOE | | **12.04** | **97.76** | **99.47** | 94.80 |
| | OE | 5000 (0.1) | 15.16 | 97.31 | 99.35 | 94.95 |
| | NTOM | | 28.91 | 84.26 | 93.40 | 94.63 |
| | POEM | | 27.46 | 86.56 | 94.68 | 94.78 |
| | MixOE | | 15.66 | 97.06 | 99.29 | **94.98** |
| | DivOE | | **14.41** | **97.43** | **99.40** | 94.81 |
| | OE | 1000 (0.02) | 16.74 | 97.11 | 99.31 | 95.03 |
| | NTOM | | 31.90 | 81.93 | 91.99 | 94.86 |
| | POEM | | 30.67 | 82.56 | 92.33 | 94.78 |
| | MixOE | | 15.80 | 97.07 | 99.29 | **95.11** |
| | DivOE | | **15.38** | **97.25** | **99.37** | 95.03 |

**Experiments on Comprehensive Explorations of the Extrapolated Data** In Tables 20 and 21, we present a more comprehensive comparison of the extrapolated outliers under the learning framework of our DivOE. To be more specific, we control the specific extrapolation ratio $r$ and the extrapolated strength $\epsilon$ to characterize the algorithm. From one perspective, we gradually increase the extrapolated strength $\epsilon$ with the fixed extrapolation ratio $r = 1$, which means we conduct the full-batch manipulation on the original surrogate OOD distribution. The results demonstrate the failure of such a overall distribution perturbation with the significant performance drop on several OOD test sets (e.g., LSUN-r and LSUN-c). Intuitively, the overall distribution manipulation for the auxiliary outliers can not perform well towards the diversification target, as it is also inconsistent with the conceptual idea of our DivOE. From the other perspective, we decrease the extrapolation ratio $r$ with the enlarged strength on informative extrapolation. Fortunately, with the diversified guidance introduced by our proposed objective in DivOE, we can find that even using $\epsilon = 0.125$ can also enhance the performance of outlier exposure with different OOD test sets, resulting in a better performance in average.

**Experiments on Using Different Synthetic Mechanism** Here, we consider some advanced techniques using Generative Adversarial Network (GAN) to synthesizing new outliers for outlier exposure. Those methods [Lee et al., 2018a, Kong and Ramanan, 2021] additionally train generative models for generating extra outliers for data augmentation, which depends on the sample quality and actually are orthogonal to our extrapolation. We conduct experiments comparing and incorporating the method [Lee et al., 2018a] for informative extrapolation in Table 12 in attached PDF. We find that the

## Table 16: Fine-grained OOD detection performance of DivOE on CIFAR-10 [Krizhevsky, 2009] as ID.

| Method | OOD dataset | | | | | | | | | | Average FPR95 |
|---|---|---|---|---|---|---|---|---|---|---|---|
| | Textures | | Places365 | | iSUN | | LSUN-r | | LSUN-c | | |
| | FPR95↓ | AUROC↑ | FPR95↓ | AUROC↑ | FPR95↓ | AUROC↑ | FPR95↓ | AUROC↑ | FPR95↓ | AUROC↑ | |
| OE | 21.88 | 96.14 | 45.22 | 91.64 | 0.00 | 100.00 | 0.00 | 100.00 | 1.02 | 99.84 | 13.62 |
| DivOE ($\epsilon = 0.01, r = 1$) | 15.32 | 97.03 | 41.27 | 91.87 | 0.00 | 100.00 | 0.00 | 100.00 | 1.31 | 99.72 | 11.58 |
| DivOE ($\epsilon = 0.01, r = 0.5$) | 14.53 | 97.19 | 44.73 | 91.31 | 0.00 | 100.00 | 0.00 | 100.00 | 2.06 | 99.63 | 12.26 |
| DivOE ($\epsilon = 0.01, r = 0.25$) | 16.72 | 96.89 | 46.08 | 91.44 | 0.00 | 100.00 | 0.00 | 100.00 | 1.31 | 99.76 | 12.82 |
| DivOE ($\epsilon = 0.01, r = 0.1$) | 17.35 | 96.91 | 45.96 | 91.63 | 0.00 | 100.00 | 0.00 | 100.00 | 1.32 | 99.79 | 12.93 |
| DivOE ($\epsilon = 0.005, r = 1$) | 16.96 | 96.89 | 43.12 | 92.06 | 0.00 | 100.00 | 0.00 | 100.00 | 0.61 | 99.88 | 12.14 |
| DivOE ($\epsilon = 0.005, r = 0.5$) | 16.85 | 96.94 | 46.57 | 91.36 | 0.00 | 100.00 | 0.00 | 100.00 | 1.07 | 99.81 | 12.90 |
| DivOE ($\epsilon = 0.005, r = 0.1$) | 19.42 | 96.64 | 46.64 | 91.64 | 0.00 | 100.00 | 0.00 | 100.00 | 0.95 | 99.84 | 13.40 |
| DivOE ($\epsilon = 0.05, r = 1$) | 27.61 | 95.64 | 53.79 | 89.86 | 2.78 | 99.55 | 1.87 | 99.68 | 24.20 | 96.44 | 22.05 |
| DivOE ($\epsilon = 0.05, r = 0.5$) | 11.85 | 97.85 | 43.34 | 91.65 | 0.02 | 99.98 | 0.02 | 99.98 | 3.21 | 99.42 | 11.69 |
| DivOE ($\epsilon = 0.05, r = 0.25$) | 12.29 | 97.76 | 42.94 | 91.88 | 0.01 | 99.99 | 0.01 | 99.99 | 2.60 | 99.54 | 11.57 |
| DivOE ($\epsilon = 0.05, r = 0.1$) | 13.45 | 97.61 | 43.27 | 91.73 | 0.00 | 99.99 | 0.00 | 99.99 | 2.09 | 99.64 | 11.76 |
| DivOE ($\epsilon = 0.125, r = 1$) | 37.58 | 94.00 | 56.40 | 89.24 | 5.59 | 99.13 | 4.17 | 99.37 | 24.54 | 96.44 | 25.66 |
| DivOE ($\epsilon = 0.125, r = 0.5$) | 14.18 | 97.50 | 44.19 | 91.79 | 0.01 | 99.99 | 0.01 | 100.00 | 1.85 | 99.69 | 12.05 |
| DivOE ($\epsilon = 0.125, r = 0.25$) | 15.92 | 97.33 | 43.76 | 92.03 | 0.00 | 100.00 | 0.00 | 100.00 | 1.39 | 99.78 | 12.21 |
| DivOE ($\epsilon = 0.125, r = 0.1$) | 15.81 | 97.39 | 44.88 | 91.82 | 0.00 | 100.00 | 0.00 | 100.00 | 1.44 | 99.77 | 12.42 |
| DivOE ($\frac{1}{4}\epsilon = 0.05, \frac{1}{4}\epsilon = 0.125$) | 12.28 | 97.73 | 44.97 | 91.51 | 0.07 | 99.98 | 0.02 | 99.99 | 3.03 | 99.47 | 12.08 |

## Table 17: Fine-grained OOD detection performance of DivOE on CIFAR-100 as ID.

| Method | OOD dataset | | | | | | | | | | Average FPR95 |
|---|---|---|---|---|---|---|---|---|---|---|---|
| | Textures | | Places365 | | iSUN | | LSUN-r | | LSUN-c | | |
| | FPR95↓ | AUROC↑ | FPR95↓ | AUROC↑ | FPR95↓ | AUROC↑ | FPR95↓ | AUROC↑ | FPR95↓ | AUROC↑ | |
| OE | 54.31 | 84.59 | 77.18 | 76.30 | 0.09 | 99.98 | 0.03 | 100.00 | 4.84 | 99.04 | 27.29 |
| DivOE ($\epsilon = 0.01, r = 1$) | 36.11 | 89.20 | 77.73 | 76.60 | 0.16 | 99.94 | 0.02 | 99.99 | 38.55 | 90.11 | 30.52 |
| DivOE ($\epsilon = 0.01, r = 0.5$) | 36.30 | 89.34 | 78.79 | 76.17 | 0.14 | 99.99 | 0.01 | 99.99 | 30.08 | 93.72 | 29.06 |
| DivOE ($\epsilon = 0.01, r = 0.25$) | 37.70 | 88.65 | 78.04 | 76.51 | 0.07 | 99.98 | 0.00 | 100.00 | 15.26 | 96.79 | 26.21 |
| DivOE ($\epsilon = 0.01, r = 0.1$) | 38.89 | 88.23 | 75.94 | 77.05 | 0.08 | 99.98 | 0.00 | 100.00 | 7.92 | 98.23 | 24.57 |
| DivOE ($\epsilon = 0.005, r = 1$) | 41.63 | 87.64 | 77.51 | 76.64 | 0.02 | 99.99 | 0.00 | 100.00 | 5.72 | 98.59 | 24.98 |
| DivOE ($\epsilon = 0.005, r = 0.5$) | 40.33 | 87.90 | 78.21 | 76.15 | 0.05 | 99.98 | 0.00 | 100.00 | 9.38 | 97.97 | 25.59 |
| DivOE ($\epsilon = 0.005, r = 0.1$) | 44.05 | 86.83 | 75.66 | 76.89 | 0.06 | 99.98 | 0.00 | 100.00 | 4.93 | 98.89 | 24.94 |
| DivOE ($\epsilon = 0.05, r = 1$) | 62.51 | 82.89 | 81.42 | 75.20 | 7.43 | 98.26 | 3.69 | 99.17 | 60.99 | 85.11 | 43.21 |
| DivOE ($\epsilon = 0.05, r = 0.5$) | 33.56 | 90.78 | 75.14 | 77.23 | 0.20 | 99.94 | 0.03 | 99.99 | 12.50 | 97.17 | 24.29 |
| DivOE ($\epsilon = 0.05, r = 0.25$) | 35.29 | 90.20 | 74.44 | 77.50 | 0.09 | 99.97 | 0.01 | 99.99 | 11.25 | 97.48 | 24.21 |
| DivOE ($\epsilon = 0.05, r = 0.1$) | 34.33 | 90.37 | 73.18 | 77.92 | 0.10 | 99.97 | 0.00 | 100.00 | 10.08 | 97.71 | 23.54 |
| DivOE ($\epsilon = 0.125, r = 1$) | 71.13 | 80.04 | 81.62 | 74.76 | 19.69 | 95.29 | 13.69 | 96.79 | 62.33 | 84.94 | 49.69 |
| DivOE ($\epsilon = 0.125, r = 0.5$) | 41.54 | 88.72 | 73.64 | 77.43 | 0.18 | 99.96 | 0.03 | 100.00 | 9.31 | 97.97 | 24.94 |
| DivOE ($\epsilon = 0.125, r = 0.25$) | 45.16 | 87.92 | 74.25 | 77.66 | 0.10 | 99.98 | 0.00 | 100.00 | 7.29 | 98.46 | 25.36 |
| DivOE ($\epsilon = 0.125, r = 0.1$) | 43.73 | 88.11 | 73.94 | 77.78 | 0.09 | 99.98 | 0.00 | 100.00 | 6.07 | 98.70 | 24.77 |
| DivOE ($\frac{1}{4}\epsilon = 0.01\ \frac{1}{4}\epsilon = 0.125$) | 31.67 | 91.12 | 76.19 | 77.03 | 0.17 | 99.95 | 0.02 | 99.99 | 24.67 | 94.73 | 26.54 |
| DivOE ($\frac{1}{4}\epsilon = 0.05, \frac{1}{4}\epsilon = 0.125$) | 34.16 | 90.61 | 74.42 | 77.53 | 0.18 | 99.94 | 0.03 | 99.99 | 12.97 | 97.10 | 24.35 |

GAN-synthesized method is hard to train under our setups. When we incorporate DivOE into the generated sample, the performance can better enhanced than the original one of GAN-based method.

**Experiments on Computational Cost** To better understand the computational budget, we conduct additional experiments and summarize the time and memory results in Table 18, which shows that DivOE can achieve better performance with comparable time and memory costs with other OE-based methods. In addition, since the extra time requirement compared with the original OE is mainly from the inner-maximization of the informative extrapolation, we also explore the effect of perturbation noise on the OOD detection performance in Table 19. The results show that we can further save extra time by reducing the outlier manipulation times with maintaining the improvement.

## Table 18: Time and memory cost of different methods.

| Dataset | Method | Time for one iteration (s) | GPU Memory (MiB) | FPR95 | AUROC | AUPR | ID-ACC |
|---|---|---|---|---|---|---|---|
| CIFAR100 | OE | **0.22** | **3769** | 27.67 | 91.89 | 97.93 | 75.41 |
| | NTOM | 0.39 | 3775 | 27.58 | 91.08 | 97.52 | 75.41 |
| | POEM | 0.79 | 3845 | 26.50 | 91.33 | 97.58 | **75.74** |
| | MixOE | 0.53 | **3769** | 35.26 | 90.31 | 97.58 | **75.74** |
| | DivOE | 0.57 | 3781 | **24.80** | **92.91** | **98.22** | 75.24 |
| CIFAR10 | OE | **0.21** | 3769 | 13.76 | 97.53 | 99.43 | 94.51 |
| | NTOM | 0.34 | 3733 | 17.76 | 91.08 | 96.40 | **95.04** |
| | POEM | 0.66 | 3835 | 17.81 | 90.96 | 96.36 | 94.90 |
| | MixOE | 0.47 | **3731** | 13.57 | 97.47 | 99.39 | 94.66 |
| | DivOE | 0.53 | 3779 | **11.66** | **97.82** | **99.48** | 94.66 |

Table 19: Perturb number and average time of the extrapolation in DivOE.

| Dataset | Method | Perturb Number | Average Time for One Iteration (s) | FPR95 | AUROC | AUPR | ID-ACC |
|---------|--------|----------------|-------------------------------------|-------|-------|------|--------|
| CIFAR100 | OE | - | 0.22 | 27.67 | 91.89 | 97.93 | 75.41 |
| | DivOE | 1 | 0.41 | 25.58 | 92.60 | 98.14 | 75.59 |
| | DivOE | 2 | 0.47 | 24.94 | 92.74 | 98.17 | 75.18 |
| | DivOE | 5 | 0.57 | 24.80 | 92.91 | 98.22 | 75.24 |
| CIFAR10 | OE | - | 0.21 | 13.76 | 97.53 | 99.43 | 94.51 |
| | DivOE | 1 | 0.38 | 12.71 | 97.70 | 99.45 | 94.78 |
| | DivOE | 2 | 0.44 | 12.10 | 97.68 | 99.45 | 95.59 |
| | DivOE | 5 | 0.53 | 11.66 | 97.82 | 99.48 | 94.66 |

**Experiments on Compatibility with other OE-based Methods** In Tables 20 and 21, we provide the fine-grained performance on OOD detection of those OE-based methods [Chen et al., 2021, Ming et al., 2022, Wang et al., 2023] with our DivOE. The overall results again confirm the effectiveness of the previously proposed OE-based methods incorporating the informative extrapolation introduced by our DivOE. Especially, DivOE via the informative extrapolation helps these methods better represent the unseen OOD inputs close to the ID decision area. The overall averaged results can refer to Table 2.

Table 20: Fine-grained results of compatibility experiments with other OE-based methods on CIFAR10 [Krizhevsky, 2009] as ID.

| Method | OOD dataset | | | | | | | | | | Average FPR95 |
|--------|----------|-------|----------|-------|-------|-------|-------|-------|-------|-------|---------------|
| | Textures | | Places365 | | iSUN | | LSUN-r | | LSUN-c | | |
| | FPR95↓ | AUROC↑ | FPR95↓ | AUROC↑ | FPR95↓ | AUROC↑ | FPR95↓ | AUROC↑ | FPR95↓ | AUROC↑ | |
| OE | 21.88 | 96.14 | 45.22 | 91.64 | 0.00 | 100.00 | 0.00 | 100.00 | 1.02 | 99.84 | 13.62 |
| NTOM | 32.78 | 86.30 | 59.03 | 67.50 | 0.00 | 100.00 | 0.00 | 100.00 | 1.17 | 99.38 | 18.59 |
| NTOM+DivOE | 14.49 | 94.03 | 53.88 | 72.31 | 0.00 | 100.00 | 0.00 | 100.00 | 5.29 | 97.53 | 14.73 |
| POEM | 34.73 | 85.28 | 57.82 | 67.93 | 0.00 | 100.00 | 0.00 | 100.00 | 0.43 | 99.74 | 18.59 |
| POEM+DivOE | 21.61 | 91.32 | 53.95 | 71.28 | 0.0 | 100.00 | 0.00 | 100.00 | 1.48 | 99.10 | 15.41 |
| MixOE | 22.23 | 95.74 | 39.45 | 92.86 | 0.26 | 99.92 | 0.27 | 99.92 | 4.40 | 99.21 | 13.32 |
| MixOE+DivOE | 18.63 | 96.34 | 42.55 | 91.68 | 0.54 | 99.86 | 0.40 | 99.86 | 9.77 | 98.34 | 14.38 |
| DOE | 21.97 | 94.87 | 15.43 | 96.74 | 0.30 | 99.89 | 0.22 | 99.91 | 1.27 | 99.67 | 7.84 |
| DOE+DivOE | 13.37 | 97.05 | 16.28 | 96.53 | 0.25 | 99.92 | 0.14 | 99.95 | 2.00 | 99.49 | 6.41 |

Table 21: Fine-grained results of compatibility experiments with other OE-based methods on CIFAR100 [Krizhevsky, 2009] as ID.

| Method | OOD dataset | | | | | | | | | | Average FPR95 |
|--------|----------|-------|----------|-------|-------|-------|-------|-------|-------|-------|---------------|
| | Textures | | Places365 | | iSUN | | LSUN-r | | LSUN-c | | |
| | FPR95↓ | AUROC↑ | FPR95↓ | AUROC↑ | FPR95↓ | AUROC↑ | FPR95↓ | AUROC↑ | FPR95↓ | AUROC↑ | |
| OE | 54.31 | 84.59 | 77.18 | 76.30 | 0.09 | 99.98 | 0.03 | 100.00 | 4.84 | 99.04 | 27.29 |
| NTOM | 59.65 | 81.76 | 80.40 | 72.46 | 0.00 | 100.00 | 0.00 | 100.00 | 2.24 | 99.46 | 28.46 |
| NTOM+DivOE | 41.65 | 88.21 | 79.45 | 72.80 | 0.00 | 100.00 | 0.00 | 100.00 | 5.04 | 98.54 | 25.23 |
| POEM | 51.72 | 84.16 | 79.94 | 72.84 | 0.00 | 100.00 | 0.00 | 100.00 | 0.34 | 99.92 | 26.40 |
| POEM+DivOE | 43.61 | 87.13 | 80.09 | 72.52 | 0.00 | 100.00 | 0.00 | 100.00 | 1.69 | 99.55 | 25.08 |
| MixOE | 61.78 | 82.13 | 72.95 | 78.20 | 2.96 | 99.32 | 1.05 | 99.75 | 33.56 | 92.88 | 34.46 |
| MixOE+DivOE | 56.91 | 83.88 | 74.26 | 77.66 | 4.00 | 99.04 | 2.02 | 99.56 | 46.42 | 89.54 | 36.72 |
| DOE | 59.92 | 83.93 | 40.50 | 91.53 | 26.37 | 95.11 | 18.35 | 96.59 | 12.90 | 97.72 | 31.61 |
| DOE+DivOE | 43.77 | 89.24 | 40.48 | 91.44 | 0.75 | 99.71 | 0.17 | 99.84 | 17.16 | 97.01 | 20.46 |

**Experiments with Limited Informative Auxiliary Outliers** In Table 22, we present the detailed performance results of the experiments with limited informative auxiliary outliers in CIFAR-10. The experiments are designed to demonstrate the consistent effectiveness of our proposed DivOE with informative extrapolation. Different from the previous sampling-based methods (e.g., NTOM [Chen et al., 2021] and POEM [Ming et al., 2022]), which assume that the potential surrogate OOD space can be arbitrarily large and cover the boundary decision area for ID and OOD inputs, our DivOE introduces the diversification target which is more general and robust given different auxiliary outliers. With such a general learning target, DivOE can perform learning and extrapolation given the surrogate OOD distribution iteratively under a single learning framework during the fine-tuning process. Empirically, it results in better performance improvement across the different setups on the auxiliary outliers in Table 20. Since the NTOM and POEM both adopt the sampling strategy to sample part of outliers (90%), we keep the same number in OE and our DivOE for the fair comparisons.

Table 22: Experiments on limited informative auxiliary outliers on CIFAR10.

| Method | Number of auxiliary outliers | FPR95↓ |
|--------|------------------------------|--------|
| OE | 50000 | 19.87 |
| | 40000 | 19.86 |
| | 30000 | 21.90 |
| | 10000 | 26.19 |
| | 5000 | 31.28 |
| NTOM | 50000 | 19.90 |
| | 40000 | 22.29 |
| | 30000 | 24.12 |
| | 10000 | 25.89 |
| | 5000 | 29.03 |
| POEM | 50000 | 20.30 |
| | 40000 | 21.39 |
| | 30000 | 21.77 |
| | 10000 | 28.10 |
| | 5000 | 27.42 |
| DivOE | 50000 | 15.95 |
| | 40000 | 15.75 |
| | 30000 | 16.82 |
| | 10000 | 17.52 |
| | 5000 | 20.96 |

**Experiments on Extrapolation via Data Augmentation**    In Tables 24, 25, 26 and 27, we replace the detailed realization of the inner-maximization of the informative extrapolation with the conventional data augmentation method, i.e., Mixup [Zhang et al., 2018], and also explore the effectiveness of combining the CutMix [Yun et al., 2019] in the learning framework. The results also confirm that the high-level idea of our DivOE is general to adopt different data manipulation or synthetic strategies to achieve the diversification target. The averaged and fine-grained performance results demonstrate the effectiveness of the proposed DivOE, which diversifies the given outlier for effective OOD detection. In addition, we also conduct experiments and combine them with the inner-maximization function. Compared with the pixel level manipulation, more coarse-grained data augmentation can also boost the performance if it can be conducted in an optimization way. If the original outliers are far away from the ID samples, data augmentation can serve as the basis for inner-maximization.

Table 23: DivOE with different inner extrapolation methods.

| Dataset | Method | Extrapolation Method | FPR95 | AUROC | AUPR | ID-ACC |
|---------|--------|----------------------|-------|-------|------|--------|
| CIFAR100 | OE | - | 27.67 | 91.89 | 97.93 | 75.41 |
| | GAN-synthesized [c] | - | 87.09 | 66.70 | 90.40 | 31.73 |
| | DivOE | PGD | **24.80** | **92.91** | **98.22** | 75.24 |
| | DivOE | Mixup | 25.70 | 92.46 | 98.07 | **76.06** |
| | DivOE | GAN-synthesized [c] | 69.69 | 81.82 | 95.16 | 29.11 |
| CIFAR10 | OE | - | 13.76 | 97.53 | 99.43 | 94.51 |
| | GAN-synthesized [c] | - | 78.75 | 72.97 | 91.88 | 50.67 |
| | DivOE | PGD | **11.66** | **97.82** | **99.48** | 94.66 |
| | DivOE | Mixup | 12.87 | 97.59 | 99.42 | **94.83** |
| | DivOE | GAN-synthesized [c] | 61.34 | 80.14 | 93.23 | 61.23 |

# D    Broader Impact

OOD detection is an important aspect of deploying reliable deep learning systems in the real world [Nguyen et al., 2015, Hendrycks et al., 2022]. It is especially important for those safety-critical applications [Bommasani et al., 2021] like financial or medical intelligence, in which a reliable

Table 24: The inner-maximization combined with different augmentation methods.

| Dataset | Method | Augmentation Method | FPR95 | AUROC | AUPR | ID-ACC |
|---------|--------|---------------------|-------|-------|------|--------|
| CIFAR100 | OE | - | 27.67 | 91.89 | 97.93 | 75.41 |
| | DivOE | Random | 26.16 | 92.39 | 98.06 | 75.56 |
| | DivOE | PGD | **24.80** | 92.91 | 98.22 | 75.24 |
| | DivOE | Mixup | 25.70 | 92.46 | 98.07 | **76.06** |
| | DivOE | CutMix | 28.46 | 91.70 | 97.89 | 75.62 |
| | DivOE | AugMix | 25.40 | 92.60 | 98.12 | 75.58 |
| | DivOE | Mixup+PGD | 25.14 | 92.79 | 98.18 | 75.78 |
| | DivOE | CutMix+PGD | 26.15 | 92.37 | 98.06 | 75.51 |
| | DivOE | AugMix+PGD | 24.92 | **93.01** | **98.24** | 75.48 |
| CIFAR10 | OE | - | 13.76 | 97.53 | 99.43 | 94.51 |
| | DivOE | Random | 13.19 | 97.60 | 99.43 | 94.81 |
| | DivOE | PGD | 11.66 | 97.82 | 99.48 | 94.66 |
| | DivOE | Mixup | 12.87 | 97.59 | 99.42 | **94.83** |
| | DivOE | CutMix | 13.95 | 97.54 | 99.42 | 94.78 |
| | DivOE | AugMix | 12.41 | 97.82 | **99.49** | 94.71 |
| | DivOE | Mixup+PGD | 12.71 | 97.70 | 99.47 | 94.68 |
| | DivOE | CutMix+PGD | 13.12 | 97.72 | 99.47 | 94.74 |
| | DivOE | AugMix+PGD | **11.49** | **97.87** | **99.49** | 94.71 |

model should have the capability to distinguish those samples having different label spaces (e.g. animals) instead of giving a prediction based on existing classes (e.g., finance products or disease). Our research work studies a general and practical research problem in outlier exposure for effective OOD detection, considering the limited informative auxiliary outliers that can not well represent the boundary outliers for the pre-trained model on the classification tasks with ID samples. It is important for effective OOD detection to gain better empirical performance through learning from the auxiliary outliers, and also conceptually figure out the critical problem in this general learning framework.

Although we take a step forward in fine-tuning with auxiliary outliers for effective OOD detection, it is not the end of this direction since there are still many practical problems to be addressed. For example, although learning with the surrogate OOD data can boost the performance of the given model in identifying the OOD inputs, fine-tuning requires extra training time and computational cost, which may bring extra resources for adjusting. Same to other OE-based methods, how to reduce the extra training cost for the model regularization is also a future direction that is worth exploring.

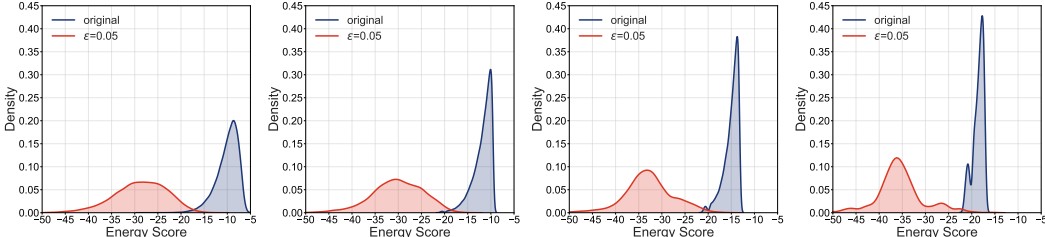

Figure 8: Verification on the effect of extrapolation via Energy Score. The density of the energy score distributions of the $\epsilon = 0.5$ extrapolated outliers under different numbers of outlier samples, 100000 for left panel, 50000 for left-middle panel, 10000 for right-middle panel, 1000 for right panel.

Table 25: Experiments of DivOE via Mixup (%). (averaged by five OOD test sets)

| $\mathcal{D}_{\text{in}}$ | Method | FPR95↓ | AUROC↑ | AUPR↑ | ID-ACC↑ |
|---|---|---|---|---|---|
| CIFAR-10 | Energy (w. $\mathcal{D}_{\text{aux}}$) | 18.69 | 90.51 | 96.15 | 94.59 |
| | Energy+Mixup (w. $\mathcal{D}_{\text{aux}}$) | 14.82 | 92.72 | 96.96 | 94.84 |
| | Energy+DivOE (Mixup) (w. $\mathcal{D}_{\text{aux}}$) | 15.02 | 92.35 | 96.78 | 94.98 |
| | Energy+CutMix (w. $\mathcal{D}_{\text{aux}}$) | 19.28 | 90.30 | 96.04 | 94.90 |
| | NTOM (with MSP) | 12.96 | 97.73 | 99.46 | 95.11 |
| | NTOM+Mixup | 12.25 | 97.85 | 99.50 | 95.04 |
| | NTOM+DivOE (Mixup) | 12.20 | 97.81 | 99.48 | 94.98 |
| | POEM | 18.32 | 90.81 | 96.27 | 94.70 |
| | POEM+Mixup | 14.70 | 92.76 | 97.00 | 94.98 |
| | POEM+DivOE (Mixup) | 15.39 | 91.89 | 96.62 | 94.77 |
| | POEM+CutMix | 17.70 | 91.25 | 96.49 | 94.88 |
| | OE | 13.62 | 97.52 | 99.41 | 94.57 |
| | OE+Mixup | 13.08 | 97.68 | 99.46 | 94.89 |
| | OE+DivOE (Mixup) | 12.87 | 97.59 | 99.42 | 94.83 |
| | OE+CutMix | 14.74 | 97.45 | 99.41 | 94.93 |
| CIFAR-100 | Energy (w. $\mathcal{D}_{\text{aux}}$) | 27.49 | 90.72 | 97.39 | 75.23 |
| | Energy+Mixup (w. $\mathcal{D}_{\text{aux}}$) | 25.27 | 92.02 | 97.81 | 75.49 |
| | Energy+DivOE (Mixup) (w. $\mathcal{D}_{\text{aux}}$) | 24.47 | 92.04 | 97.79 | 75.53 |
| | Energy+CutMix (w. $\mathcal{D}_{\text{aux}}$) | 29.87 | 89.72 | 97.13 | 75.59 |
| | NTOM (with MSP) | 26.13 | 92.32 | 98.07 | 75.54 |
| | NTOM+Mixup | 25.41 | 92.75 | 98.18 | 75.60 |
| | NTOM+DivOE (Mixup) | 24.44 | 92.73 | 98.15 | 75.84 |
| | POEM | 25.57 | 91.73 | 97.69 | 76.01 |
| | POEM+Mixup | 24.46 | 92.19 | 97.84 | 75.57 |
| | POEM+DivOE (Mixup) | 24.14 | 92.16 | 97.81 | 75.57 |
| | POEM+CutMix | 28.23 | 90.75 | 97.41 | 75.90 |
| | OE | 27.29 | 91.98 | 97.96 | 75.51 |
| | OE+Mixup | 27.20 | 92.22 | 98.03 | 75.87 |
| | OE+DivOE (Mixup) | 25.70 | 92.46 | 98.07 | 76.06 |
| | OE+CutMix | 31.14 | 90.98 | 97.70 | 75.66 |

Table 26: OOD detection performance on CIFAR-10 [Krizhevsky, 2009] as ID. All methods are trained on the same backbone. Values are percentages. ↑ indicates larger values are better, and ↓ indicates smaller values are better. (averaged by multiple trials)

| Method | OOD dataset | | | | | | | | | | ID ACC |
|---|---|---|---|---|---|---|---|---|---|---|---|
| | Textures | | Places365 | | iSUN | | LSUN-r | | LSUN-c | | |
| | FPR95↓ | AUROC↑ | FPR95↓ | AUROC↑ | FPR95↓ | AUROC↑ | FPR95↓ | AUROC↑ | FPR95↓ | AUROC↑ | |
| Energy (w. $\mathcal{D}_{\text{aux}}$) | 34.83 | 84.76 | 58.34 | 67.95 | 0.00 | 100.00 | 0.00 | 100.00 | 0.28 | 99.84 | 94.59 |
| Energy+Mixup (w. $\mathcal{D}_{\text{aux}}$) | 20.03 | 90.84 | 48.51 | 75.38 | 0.00 | 100.00 | 0.00 | 100.00 | 5.55 | 97.38 | 94.84 |
| Energy+DivOE (Mixup) (w. $\mathcal{D}_{\text{aux}}$) | 23.25 | 89.57 | 51.50 | 72.39 | 0.00 | 100.00 | 0.00 | 100.00 | 0.38 | 99.80 | 94.98 |
| NTOM(with MSP) | 21.04 | 96.31 | 43.27 | 92.43 | 0.00 | 100.00 | 0.00 | 100.00 | 0.52 | 99.92 | 95.11 |
| NTOM+Mixup | 17.98 | 96.83 | 42.13 | 92.65 | 0.00 | 100.00 | 0.00 | 100.00 | 1.12 | 99.79 | 95.04 |
| NTOM+DivOE (Mixup) | 18.12 | 96.76 | 42.16 | 92.40 | 0.00 | 100.00 | 0.00 | 100.00 | 0.71 | 99.88 | 94.98 |
| POEM | 34.52 | 84.54 | 57.00 | 69.55 | 0.00 | 100.00 | 0.00 | 100.00 | 0.09 | 99.96 | 94.70 |
| POEM+Mixup | 21.19 | 89.99 | 48.59 | 75.45 | 0.00 | 100.00 | 0.00 | 100.00 | 3.70 | 98.35 | 94.98 |
| POEM+DivOE (Mixup) | 22.36 | 89.03 | 54.29 | 70.61 | 0.00 | 100.00 | 0.00 | 100.00 | 0.32 | 99.83 | 94.77 |
| OE | 21.88 | 96.14 | 45.22 | 91.64 | 0.00 | 100.00 | 0.00 | 100.00 | 1.02 | 99.84 | 94.57 |
| OE+Mixup | 19.01 | 96.68 | 44.70 | 91.97 | 0.00 | 100.00 | 0.03 | 100.00 | 1.67 | 99.74 | 94.89 |
| OE+DivOE (Mixup) | 19.92 | 96.40 | 43.59 | 91.69 | 0.00 | 100.00 | 0.00 | 100.00 | 0.82 | 99.86 | 94.83 |

Table 27: OOD detection performance on CIFAR-100 [Krizhevsky, 2009] as ID. All methods are trained on the same backbone. Values are percentages. ↑ indicates larger values are better, and ↓ indicates smaller values are better. (averaged by multiple trials)

| Method | OOD dataset | | | | | | | | | | ID ACC |
| | Textures | | Places365 | | iSUN | | LSUN-r | | LSUN-c | | |
| | FPR95↓ | AUROC↑ | FPR95↓ | AUROC↑ | FPR95↓ | AUROC↑ | FPR95↓ | AUROC↑ | FPR95↓ | AUROC↑ | |
|---|---|---|---|---|---|---|---|---|---|---|---|
| Energy (w. $\mathcal{D}_{aux}$) | 57.45 | 80.77 | 79.60 | 72.95 | 0.00 | 100.00 | 0.00 | 100.00 | 0.41 | 99.90 | 75.23 |
| Energy+Mixup (w. $\mathcal{D}_{aux}$) | 42.00 | 87.24 | 74.82 | 75.16 | 0.00 | 100.00 | 0.00 | 100.00 | 9.53 | 97.69 | 75.49 |
| Energy+DivOE (Mixup) (w. $\mathcal{D}_{aux}$) | 44.64 | 86.13 | 77.01 | 74.28 | 0.00 | 100.00 | 0.00 | 100.00 | 0.72 | 99.80 | 75.53 |
| NTOM(with MSP) | 52.99 | 85.03 | 75.53 | 77.08 | 0.01 | 100.00 | 0.00 | 100.00 | 2.14 | 99.52 | 75.54 |
| NTOM+Mixup | 42.36 | 88.06 | 74.07 | 77.89 | 0.09 | 99.98 | 0.00 | 100.00 | 10.53 | 97.84 | 75.60 |
| NTOM+DivOE (Mixup) | 44.37 | 87.21 | 74.32 | 77.20 | 0.02 | 99.99 | 0.01 | 100.00 | 3.50 | 99.23 | 75.84 |
| POEM | 51.55 | 83.56 | 76.23 | 75.10 | 0.00 | 100.00 | 0.00 | 100.00 | 0.09 | 99.98 | 76.01 |
| POEM+Mixup | 39.70 | 88.09 | 77.06 | 74.39 | 0.00 | 100.00 | 0.00 | 100.00 | 5.53 | 98.47 | 75.57 |
| POEM+DivOE (Mixup) | 43.30 | 86.49 | 77.12 | 74.43 | 0.00 | 100.00 | 0.00 | 100.00 | 0.28 | 99.90 | 75.57 |
| OE | 54.31 | 84.59 | 77.18 | 76.30 | 0.09 | 99.98 | 0.03 | 100.00 | 4.84 | 99.04 | 75.51 |
| OE+Mixup | 47.11 | 86.53 | 74.27 | 77.66 | 0.22 | 99.93 | 0.09 | 99.99 | 14.29 | 97.00 | 75.87 |
| OE+DivOE (Mixup) | 48.35 | 86.07 | 75.04 | 77.33 | 0.13 | 99.97 | 0.02 | 100.00 | 4.97 | 98.94 | 76.06 |

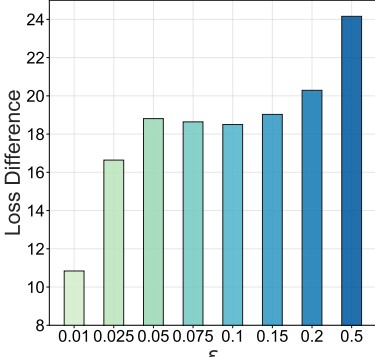

Figure 9: OOD loss difference (i.e., the maximization part in Eq. (4)) between original outliers and the extrapolated outliers with different $\epsilon$ in DivOE.

| $\epsilon$ | Loss difference |
|---|---|
| 0.01 | 10.84 |
| 0.025 | 16.64 |
| 0.05 | 18.81 |
| 0.075 | 18.64 |
| 0.1 | 18.50 |
| 0.15 | 19.03 |
| 0.2 | 20.29 |
| 0.5 | 24.16 |

Table 28: OOD loss difference (i.e., the maximization part in Eq. (4)) between original outliers and the extrapolated outliers with different $\epsilon$ in DivOE.

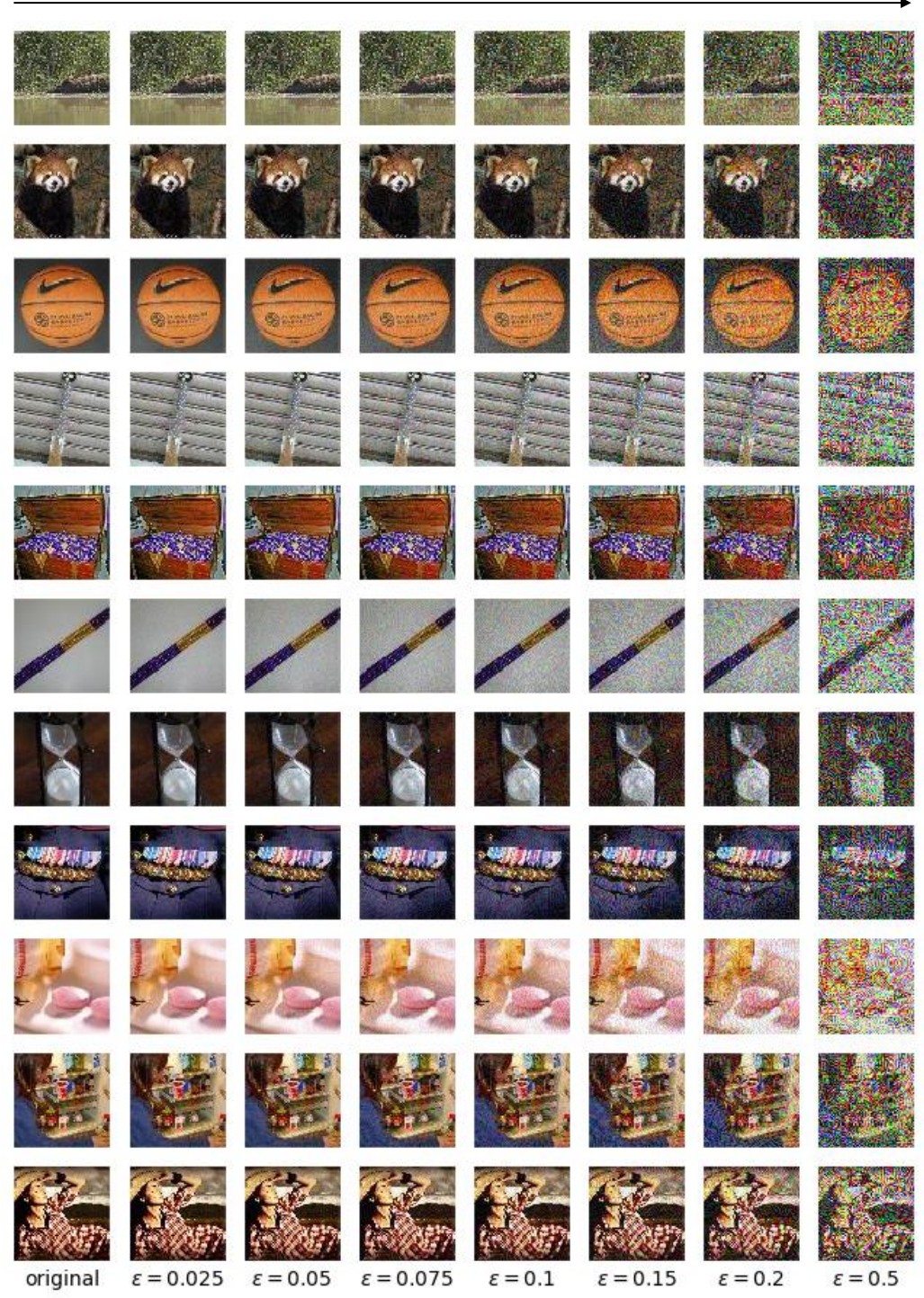

original    $\varepsilon = 0.025$    $\varepsilon = 0.05$    $\varepsilon = 0.075$    $\varepsilon = 0.1$    $\varepsilon = 0.15$    $\varepsilon = 0.2$    $\varepsilon = 0.5$

Figure 10: Examples of the extrapolated auxiliary outliers by controlling the $\epsilon$ in Eq. (10) from 0 (i.e., original) to 0.5. By different levels of pixel manipulation, the extrapolated samples can cover more potential decision areas for OOD inputs.

