# OpenReview forum: "Diversified Outlier Exposure for Out-of-Distribution Detection via Informative Extrapolation"
_NeurIPS.cc/2023/Conference — NeurIPS 2023 poster_

### Official Review · Reviewer_Mh63 · 2023-07-04

**Soundness:** 3 good
**Presentation:** 3 good
**Contribution:** 3 good
**Rating:** 5
**Confidence:** 4

**Summary:**

The authors of this paper present an innovative methodology for out-of-distribution (OOD) detection. While existing methodologies typically involve the direct use of given OOD samples, this paper introduces a new approach that applies perturbations to OOD samples, allowing the model to experience a more diverse set of OOD samples during training.

The authors define these perturbations using an adversarial loss based on the uniform distribution loss commonly applied to OOD samples. This creates directional guidance for each instance, determining the direction of the gradient and thus defining the perturbations applied to OOD samples.

The results demonstrate that using perturbed OOD samples improves the performance of OOD detection across several key metrics, including False Positive Rate at 95% Recall (FPR95), Area Under the Receiver Operating Characteristic Curve (AUROC), and Area Under the Precision-Recall Curve (AUPR), when compared to existing methodologies.

**Strengths:**

This paper's novel approach of applying perturbations to a given data instance in order to utilize a broader array of samples and those tightly located at the decision boundary is indeed a reasonable and intriguing choice. The authors' concept of adversariality against the uniform loss, which implies a concentration of prediction towards a particular class, exhibits an interesting property worth exploring further.

Experimentally, the authors provide a substantial ablation study on the hyperparameters used in loss composition, which adds to the comprehensiveness of their methodology. Notably, the use of t-SNE based plotting allows for a clear visualization of how the OOD samples are perturbed to be situated very closely to in-class samples. It also arise the question of "how would ood behave for the different perturbation types?".

**Weaknesses:**

The authors have presented an interesting methodology that applies adversarial perturbations based on a particular loss function to enhance the performance of out-of-distribution detection. However, it would be important for the authors to provide empirical evidence that demonstrates the superiority of this adversarial perturbation over other types of perturbations. This would require a systematic and rigorous experimentation design and would ideally be conducted across various datasets and under different conditions to ensure the results are robust and generalizable.

In the final implementation of the authors' methodology, it's noted that both the perturbed and unperturbed out-of-distribution (OOD) samples are simultaneously considered in the loss function. However, the manuscript doesn't sufficiently explain the rationale behind this particular choice. It would be particularly interesting to see a comparison of results when using only perturbed samples, only unperturbed samples, or both, in the loss function.

a crucial aspect that needs further clarification and discussion is the ratio of in-distribution to OOD samples used in the training process. The choice of this ratio can significantly affect the performance and robustness of the model. For instance, a too high proportion of OOD samples could make the model overly sensitive to outliers, while a too low proportion might not adequately expose the model to OOD scenarios.

While the paper shows promising results in the specific contexts tested, it would be beneficial for the authors to provide a more extensive analysis covering a range of different OOD datasets. If the authors can provide empirical evidence demonstrating the consistent performance of their method across diverse OOD datasets, it would significantly strengthen their claims.

it's unclear from the manuscript how each OOD sample is directed towards a specific class during this adversarial perturbation process. Specifically, they should explain how the gradient direction, which determines the perturbations applied to the OOD samples, correlates with the movement of these samples towards specific classes.

I am more than eager to increase my score if the questions above are adequately answered.



**Questions:**

Please see the section weaknesses.

**Limitations:**

Please see the section weaknesses.

---

> ### Author Rebuttal · Authors · 2023-08-08
>
> Thank you for your time devoted to reviewing this paper and your constructive suggestions. Here are our detailed replies to your questions.
>
> > **Q1:** However, it would be important for the authors to provide empirical evidence that demonstrates the superiority of this adversarial perturbation over other types of perturbations.....
>
> Thanks for the constructive comments. We would like to clarify that **DivOE was proposed as a general framework** for diversifying outlier exposure with **partially conduct informative extrapolation**, which allows any appropriate noise or augmentation techniques to be incorporated.
>
> To show the effectiveness of PGD, we conduct the following experiments. On the one hand, we compare the random noise, re-adversarial perturbation, and adversarial perturbation to justify its rationality. The results are summarized in the Table 15 in attached PDF and show **the adversarial perturbation achieve better performance than other types of perturbations**. On the other hand, we compare with other adversarial attacks and data augmentation methods for inner-maximization part in Table 4 in attached PDF and show that **PGD with the uniform distribution loss is generally a better way** to achieve higher performance due to the optimization originality. We will add the complete results with the corresponding discussion in our draft.
>
> > **Q2:** ...However, the manuscript doesn't sufficiently explain the rationale behind this particular choice...
>
> Considering the initial motivation to diversify outliers, we explore to extrapolate more potential OOD distribution beyond the original auxiliary ones. For the unperturbed OOD samples used in training, we hope **the training can still learn the knowledge from the original auxiliary outliers**, for the perturbed OOD samples, we hope **the training can find different but informative outliers beyond the original ones** to improve the generalization of outlier exposure.
>
> Formally, Eq.(5) is an instance-level realization of **Eq.(4) that actually contains the perturbed part and the unperturbed part**. The deduction from Eq.(4) to Eq.(5) requires to merge the same terms in Eq.(4) first, and then specifically set $\beta=\frac{n-rn}{n}$ with a subsequent rescaling. Finally, we have the formulation that a portion of data to conduct maximization part for diversifying the existing data distribution and the remaining other portion of data corresponds to learning on the original outliers. We will add more explanation about the final implementation.
>
> To better understand the specific effects  of different samples, it can be referred to the left panel of Figure 4 (in the original submission), in which we show the OOD detection performance of DivOE under different portions of perturbed data during training. **Generally, using a small part of perturbed samples can achieve the extrapolation target and improve the performance**. However, if we totally perturb all the data, it is even worse than the original OE.
>
> > **Q3:** a crucial aspect that needs further clarification and discussion is the ratio of in-distribution to OOD samples used in the training process...
>
> Thanks for pointing out this. We generally **keep the same auxiliary outlier across different methods for fair comparison**. For the effects under different ratios of auxiliary outliers to ID samples, we conduct the extra experiments in Table 11 in attached PDF to show the results. Specifically, we vary the outlier ratio from 1.0 (50000) to 0.02 (1000) and keep the ID data number=50000 unchanged. Generally, the empirical results show that all the OE-based methods can achieve better OOD detection performance, and **our proposed DivOE can consistently improve the performance under different ratios. In comparison, with less auxiliary outliers, the model performance of ID classification will be less affected**. We will add the complete results with the corresponding discussion in our draft.
>
> > **Q4:** it would be beneficial for the authors to provide a more extensive analysis covering a range of different OOD datasets...
>
> We would like to re-explain that all the OOD detection results in Table 1 and Table 2 are averaged on five different OOD test datasets, i.e., Textures, Places365, iSUN, LSUN-r, LSUN-c, which are also benchmarked OOD test sets used in previous works. For the direct comparison under those OOD test sets, we list the performance of our DivOE with the original OE in Table 16 in attached PDF, which show that **DivOE can generally perform better across different OOD test sets**. For more results on other OE-based methods, due to the space limit, we kindly refer the reviewer to Tables 4-8 in our Appendix for more results and analysis.
>
> > **Q5:** how each OOD sample is directed towards a specific class during this adversarial perturbation process...
>
> Due to the space limit, we place the answer to the general response.

---

> ### Author Response · Authors · 2023-08-14
> **[Invitation to rolling discussion] Need further clarification?**
>
> Dear Reviewer,
>
> Thanks very much for your constructive comments on our work. We have tried our best to address the concerns. Is there any unclear point so that we should/could further clarify?
>
> Best regards.

---

> > ### Comment · Reviewer_Mh63 · 2023-08-15
> > **Thanks for your detailed response.**
> >
> > Most of my concerns have been resolved by the author's response. I am willing to increase the score to 5.

---

> > > ### Author Response · Authors · 2023-08-15
> > > **Thanks for your positive response!**
> > >
> > > Thank you very much for the positive feedback! We are glad to hear that our response solved your concerns and will incorporate all suggestions in the revision.

---

### Official Review · Reviewer_1Wzd · 2023-07-04

**Soundness:** 3 good
**Presentation:** 3 good
**Contribution:** 2 fair
**Rating:** 6
**Confidence:** 4

**Summary:**

The manuscript studies image-wide OOD detection in presence of auxiliary negative data. The negative data is often limited and therefore cannot fully encompass the distribution of inliers. Consequently, contemporary learning procedures fail to deliver classifiers resilient to outliers. To overcome this issue, the manuscript presents a method for extrapolating the negative data towards all modes of the inlier distribution. The proposed method first calculates the gradient of arbitrary OOD score over the input. Then, the sign of the gradient is used to direct the negative input samples towards the inlier distribution. The final learning algorithm uses both initial and extrapolated auxiliary negatives to train the classifier resilient to outliers. The proposed method outperforms relevant related works on small image benchmarks.

**Strengths:**

S1. The manuscript deals with an important issue.

S2. Extrapolation of auxiliary negative data towards modes of inlier distribution intuitively makes sense.

S3. The developed method achieves competitive results on considered benchmarks.

S4. The developed method can be combined with existing OOD detectors (e.g. Energy, MSP, ... )

**Weaknesses:**

W1.  The manuscript does not discuss the effectiveness of the method when there is only a small auxiliary dataset available. It seems that the developed method still requires a broad OE dataset (Tiny-ImageNet as stated in L236).

W2. The manuscript does not consider relevant related works which use synthetic negatives created by generative models [a,b,c]. Synthetic negatives are an effective way for augmenting the auxiliary dataset and the proposed method should outperform methods trained on a mixture of real and synthetic negative data.

W4. The manuscript does not reflect on the additional computational budget (time and memory) required by the method over the OE baseline.

[a] Shu Kong, Deva Ramanan: OpenGAN: Open-Set Recognition via Open Data Generation. ICCV 2021

[b] Matej Grcic, Petra Bevandic, Sinisa Segvic: Dense Open-set Recognition with Synthetic Outliers Generated by Real NVP. VISAPP 2021.

[c] Kimin Lee, Honglak Lee, Kibok Lee, Jinwoo Shin: Training Confidence-calibrated Classifiers for Detecting Out-of-Distribution Samples. ICLR 2018.

**Questions:**

C1. Can the proposed method work on large-scale experiments [d]

C2. Is the extrapolation of outlier data towards inliers always possible or there are some requirements that should be met?
Analysis similar to [e] could improve the manuscript.

[d] Haoqi Wang, Zhizhong Li, Litong Feng, Wayne Zhang:
ViM: Out-Of-Distribution with Virtual-logit Matching. CVPR 2022.

[e] Zhen Fang, Yixuan Li, Jie Lu, Jiahua Dong, Bo Han, Feng Liu:
Is Out-of-Distribution Detection Learnable? NeurIPS 2022

**Limitations:**

Although promised in Appendix D (L435), the limitations are not clearly stated. One possible limitation might be W1.

---

> ### Author Rebuttal · Authors · 2023-08-08
>
> Thank you for your time devoted to reviewing this paper and your constructive suggestions. Here are our detailed replies to your questions.
>
> > **W1:** The manuscript does not discuss the effectiveness of the method when there is only a small auxiliary dataset available. It seems that the developed method still requires a broad OE dataset (Tiny-ImageNet as stated in L236).
>
> Here, we added the experiments under different numbers of auxiliary outliers used for all OE-based methods. The experimental results are summarized in Table 11 in attached PDF, which indicates that the performance of OOD detection can be generally better if we use more outliers. **Under smaller number of samples, i.e., from 50000 to 1000 (relatively smaller than 50000 ID data), DivOE can consistently perform better than OE**.
>
> > **W2:** The manuscript does not consider relevant related works which use synthetic negatives created by generative models [a,b,c]. Synthetic negatives are an effective way for augmenting the auxiliary dataset and the proposed method should outperform methods trained on a mixture of real and synthetic negative data.
>
> Thanks for recommending these related works [a,b,c], and we will add the proper discussion and comparison in the revision. Those methods additionally train generative models for generating extra outliers for data augmentation, which **depends on the sample quality and actually are orthogonal to our extrapolation**. We conduct experiments comparing and incorporating the method [c] for informative extrapolation in Table 12 in attached PDF. We find that the GAN-synthesized method is hard to train under our setups. **When we incorporate DivOE into the generated sample, the performance can better enhanced than the original performance of GAN-based method**. More results about [a,b,c] will be conducted and added to the submission.
>
> [a] Shu Kong, Deva Ramanan: OpenGAN: Open-Set Recognition via Open Data Generation. ICCV 2021
>
> [b] Matej Grcic, Petra Bevandic, Sinisa Segvic: Dense Open-set Recognition with Synthetic Outliers Generated by Real NVP. VISAPP 2021.
>
> [c] Kimin Lee, Honglak Lee, Kibok Lee, Jinwoo Shin: Training Confidence-calibrated Classifiers for Detecting Out-of-Distribution Samples. ICLR 2018.
>
> > **Actually W3:** The manuscript does not reflect on the additional computational budget (time and memory) required by the method over the OE baseline.
>
> Thanks for the constructive comments. To better understand the computational budget, we conduct additional experiments and summarize the time and memory results in Table 13 in attached PDF, which shows that **DivOE can achieve better performance with comparable time and memory costs with other OE-based methods**.
>
> In addition, since the extra time requirement compared with the original OE is mainly from the inner-maximization of the informative extrapolation, we also explore the effect of perturbation noise on the OOD detection performance in Table 14 in attached PDF. The results show that **we can further save extra time by reducing the outlier manipulation times with maintaining the improvement**.
>
> We will update the complete results and corresponding discussion in our draft.
>
> > **Q1:** Can the proposed method work on large-scale experiments
>
> Thanks for the question. We added additional experiments on ImageNet (with ImageNet 21k as the auxiliary outlier dataset), and present the results in Table 6 in attached PDF. **The results demonstrate the consistent effectiveness of DivOE on the large-scale data**.
>
> We will add the complete results with the corresponding discussion in our submission.
>
> > **Q2:** Is the extrapolation of outlier data towards inliers always possible or there are some requirements that should be met? Analysis similar to [e] could improve the manuscript.
>
> Thanks for the constructive question and insightful suggestion! The extrapolation of outlier data towards inliers **may not always be possible**. Similar to the condition analysis in [e], the informative extrapolation that targets better characterize the decision boundary between ID and OOD also needs some requirements or realization assumptions. Intuitively, it mainly **relies on auxiliary outliers and specific extrapolation techniques**. For the former, the original outliers **should not be far away from the ID sample**; For the latter, the extrapolation techniques should be able to **add enough perturbation or semantic changes to the original outlier**. We will extend the above discussion with more conditional derivation by referring to the analysis in [e].
>
> [e] Zhen Fang, Yixuan Li, Jie Lu, Jiahua Dong, Bo Han, Feng Liu: Is Out-of-Distribution Detection Learnable? NeurIPS 2022

---

> > ### Comment · Reviewer_1Wzd · 2023-08-16
> > **Post Rebuttal**
> >
> > The authors thoroughly resolved my concerns hence I increase my score.

---

> > > ### Author Response · Authors · 2023-08-16
> > > **Thanks for the post-rebuttal response!**
> > >
> > > Thank you very much for the post-rebuttal response and positive support! We will incorporate all suggestions in the revision.

---

### Official Review · Reviewer_vD9G · 2023-07-05

**Soundness:** 3 good
**Presentation:** 3 good
**Contribution:** 3 good
**Rating:** 6
**Confidence:** 2

**Summary:**

The paper tries to solve the problem that sampled auxiliary informative outliers may not be sufficient and diverse enough to recover the data boundary in the OOD detection setting. To achieve this, the authors propose a new learning objective with information extrapolation, where second term expands the surrogate OOD distributions towards a more diversified one. The authors have provided theoretical analysis to show that for Gaussian mixture model and binary classification problem, DivOE can extrapolate the
outlier boundary towards ID data. In the experiments, the authors adopt image ID datasets such as CIFAR10, CIFAR100, and results show that DivOE achieves better performances over several evaluation metrics.

**Strengths:**

S1. The paper targets an important research problem within the OOD detection research community, and proposes a new auxiliary outlier generation and learning objective to target the research problem that surrogate auxiliary outliers are not sufficient and diverse enough.

S2. The learning objective is simple but adaptable to different post-hoc scoring functions, augmentation techniques and sampling techniques to auxiliary outliers.

S3. The paper has provided a solid theoretical analysis that shows the effectiveness of DivOE within a simplified binary classification setting. The result of Theorem 3.1 is consistent with the authors' explanations and observations in the previous sections.

S4. The paper provides good experiment comparison to exciting methods. They have used several evaluation metrics to demonstrate the effectiveness of the approach.

S5. The writing is very clear and presentation is easy to understand.

**Weaknesses:**

W1. In Theorem 3.1, The hypothesis class of the binary classification problem is overly simplified to only linear decision boundary. It would be nicer if the authors can provide theoretical results that generalize to more complex hypothesis class.

W2. As Figure 4 shows, the extrapolation ratio and diversified strength are two variables that affect the OOD detection results. If extrapolation ratio is between 0.9~1.0, the performance may degrade and become worse than OE. However, the authors have not mentioned any general guidance to tune those two variables, especially for unseen OOD detection problems.

W3. The authors should consider to include a table comparing the number of outliers generated by DivOE and other benchmark methods in order to achieve similar detection performance scores during experiments.

W4. The authors should also include bold numbers for the best performing algorithms for Table 11, Table 12 and Table 13.

**Questions:**

Q1. The experiments are only performed on two simple image classification tasks with CIFAR10 and CIFAR100. The reviewer is wondering whether the authors have used other image datasets or tabular datasets for evaluation.

Q2. Is there a general guideline in how to choose the extrapolation ratio, augmentation techniques, diversified strength, OOD scores when using DivOE on unseen OOD detection task? It seems that the four factors play an important role in detection performance.

Q3. How would $\eta$ and $\alpha$ play an effect in the training steps of DivOE?

Q4. Could different data augmentation techniques be used in combination with the inner-maximization function? Would the augmentation function boost up the performance?

**Limitations:**

The authors have adequately addressed the limitations and potential negative societal impact of their work, as listed in the NeurIPS checklist. The reviewer would appreciate if the authors can address the question and weakness section.

---

> ### Author Rebuttal · Authors · 2023-08-08
>
> Thank you for your time devoted to reviewing this paper and your constructive suggestions. Here are our detailed replies to your questions.
>
> > **W1:** About Theorem 3.1
>
> Thanks for the constructive comments. In this work, we follow POEM [Ming et al., 2022] to provide an explanation about DivOE under the linear case. We will generalize the theoretical analysis into the non-linear counterpart by combining the theoretical advancement of deep learning.
>
> > **W2, Q2:** About guidance for extrapolation factors
>
> Thanks for the constructive question. Intuitively, we hope the extrapolation can realize the diversification target and cover more potential OOD distributions beyond the original auxiliary ones. Under such high-level guidance, generally, the extrapolation ratio **is expected to be not too large** to totally change the whole original auxiliary distribution, which is empirically demonstrated to be negative for OE, as Figure 4(a) shows. As for diversified strength, we kindly refer the reviewer to the left middle panel of Figure 3, which serves as a search space for synthesizing different outliers using different $\epsilon$.
>
> In practical hyper-parameter tuning, we may generally suggest starting with the original OE baseline (i.e., extrapolation ratio equals 0 and diversified strength equals 0) and **enlarging the extrapolation ratio or diversified strength step by step, finding a stage where the OOD detection performance on the validation set will not be further enhanced**. For the coarse-grained augmentations and specific OOD scores, it may be further decided by other conditions on data quality. We will add these discussions to our submission for clarity.
>
> > **W3:** The authors should consider to include a table comparing the number of outliers generated by DivOE and other benchmark methods in order to achieve similar detection performance scores during experiments.
>
> **It may be hard to align the sample number as different methods have different synthesis motivations and but we can actually present the number of generated samples**. To facilitate the comparison, we summarize the specific outlier numbers that were manipulated in the adopted algorithm with its performance in OOD detection in Table 8 in attached PDF. The results show that, compared with previous methods, either need go through all the outliers for sampling or change the training on all the auxiliary outliers, our DivOE can conduct extrapolation on a small portion of original outliers but achieve generally better performance across different ID classification tasks.
>
> > **W4:** The authors should also include bold numbers for the best performing algorithms for Table 11, Table 12 and Table 13.
>
> We will revise Tables 11, 12, and 13 to bold the numbers of the best-performing algorithms.
>
> > **Q1:** The experiments are only performed on two simple image classification tasks with CIFAR10 and CIFAR100. The reviewer is wondering whether the authors have used other image datasets or tabular datasets for evaluation.
>
> Thanks for the question. We accordingly conduct **the experiments on ImageNet, and adopt ImageNet 21k as auxiliary outliers**. We summarize the results in Table 6 in attached PDF, and the results again confirm that our DivOE achieves a competitive performance than other counterparts. We will update all the complete results with the corresponding discussion in our draft.
>
> > **Q3:** How would $\eta$ and $\alpha$ play an effect in the training steps of DivOE?
>
> Thanks for the question. If we understand correctly, $\eta$ in our algorithm indicates the learning rate, which we **kept the same** across different OE methods. As for $\alpha$, it is a step size adopted in the informative extrapolation. To better understand its effect, we conducted the experiments by changing the step size in the PGD attack in Table 9 in attached PDF. Since the step size is only related to the optimization of the constrained maximization, **the effects on the final performance of DivOE are not significant**. By enlarging the step size, we may save more extra time on the inner maximization to reduce the time cost relatively, which is beneficial for algorithm deployment.
>
> > **Q4:** Could different data augmentation techniques be used in combination with the inner-maximization function? Would the augmentation function boost up the performance?
>
> For different data augmentation techniques, we accordingly conduct experiments and combine them with the inner-maximization function in Table 10 in attached PDF. First, we simply replace the PGD attack-based noise with other data augmentation methods. It seems that in our experiments, **those augmentation methods can improve the performance of the original OE. In comparison, PGD can achieve the better maximization target for extrapolation**. Compared with the pixel level manipulation, more coarse-grained data augmentation can also boost the performance if it can be conducted in an optimization way. In addition, if the original outliers are far away from the ID samples, data augmentation can serve as the basis for inner-maximization.

---

> > ### Comment · Reviewer_vD9G · 2023-08-14
> > **Response to the Authors**
> >
> > The authors have addressed my concerns very thoroughly and have provided detailed experiments to support their claims. While the reviewer is still concerned about Theorem 3.1 (if it is generalizable to non-linear classifiers) and hyper-parameter tuning for extrapolation factors (lack of experiments/ablation studies for this), the reviewer would keep the current score and recommend acceptance of the paper.

---

> > > ### Author Response · Authors · 2023-08-15
> > > **Thank you for the positive support!**
> > >
> > > Many thanks for the positive support and we will consider all suggestions in the revision.
> > >
> > > Regarding the non-linear extension of theoretical analysis, we would like to kindly clarify that the key point lies in changing the basis of linear hypothesis for the data property analysis in sample complexity, which requires to solve the non-trival inequality scaling under the non-linear cases. It is not very straightforward and current advances on sample complexity cannot be applied in the extension. We are still tracing more studies for deduction and will update this part once we find the proper theoretical tools.
> > >
> > > For four extrapolation factors, we summarize the results in the following and discuss the general guidance for tuning. First, MSP score and PGD are empirically demonstrated to be the appropriate choice for extrapolation, which can be fixed (Please refer to Tables 1 and 2). Second, extrapolation ratio and strength (i.e., $\epsilon$) have general trends either individually or jointly considering them. **The performance of the extrapolation ratio is monotonous regarding different $\epsilon$.** The large values would degrade performance due to the loss of diversity on the auxiliary outliers. **Then, the $\epsilon$ could be tuned after the other factors are set to appropriate choices.** Our empirical results show that very small or large epsilon should be avoided, and epsilons from 0.05 to 0.1 can generally achieve satisfactory results (Please refer to Table 3).
> > >
> > > **Table 1.** DivOE with different OOD scores.
> > >
> > > | Dataset | Method | OOD Score for Extrapolation | FPR95 |
> > > | :------- | :----- | :-----------: | :---: |
> > > | CIFAR100 | OE | - | 27.67 |
> > > | | DivOE | MSP | 24.80 |
> > > | | DivOE | Energy | 25.45 |
> > > | CIFAR10 | OE | - | 13.76 |
> > > | | DivOE | MSP | 11.66 |
> > > | | DivOE | Energy | 13.33 |
> > >
> > > **Table 2.** DivOE with different extrapolation techniques.
> > >
> > > | Dataset  | Method | Augmentation Technique | FPR95 |
> > > | :------- | :----- | :-----------: | :---: |
> > > | CIFAR100 | OE     |       -       | 27.67 |
> > > |          | DivOE  |    Random     | 26.16 |
> > > |          | DivOE  |     FGSM      | 25.58 |
> > > |          | DivOE  |      PGD      | **24.80** |
> > > | | DivOE | Mixup | 25.70 |
> > > | | DivOE | CutMix | 28.46 |
> > > | | DivOE | AugMix | 25.40 |
> > > | CIFAR10  | OE     |       -       | 13.76 |
> > > |          | DivOE  |    Random     | 13.19 |
> > > |          | DivOE  |     FGSM      | 12.71 |
> > > |          | DivOE  |      PGD      | **11.66** |
> > > | | DivOE | Mixup | 12.87 |
> > > | | DivOE | CutMix | 13.95 |
> > > | | DivOE | AugMix | 12.41 |
> > >
> > > **Table 3.** DivOE with different extrapolation ratios and strengths.
> > >
> > > | Dataset  | Method | Extrapolation ratio $r$ | Extrapolation strength $\epsilon$ | FPR95 |
> > > | :------- | :----- | :-------: | :-------: | :---: |
> > > | CIFAR100 | OE | - | - | 27.67 |
> > > | | DivOE | 1 | 0.01 | 30.52 |
> > > | | DivOE | 0.5 | 0.01 | 29.06 |
> > > | | DivOE | 0.25 | 0.01 | 26.21 |
> > > | | DivOE | 0.1 | 0.01 | 24.57 |
> > > | | DivOE | 1 | 0.05 | 43.21 |
> > > | | DivOE | 0.5 | 0.05 | 24.80 |
> > > | | DivOE | 0.25 | 0.05 | 24.21 |
> > > | | DivOE | 0.1 | 0.05 | 23.54 |
> > > | | DivOE | 1 | 0.125 | 49.69 |
> > > | | DivOE | 0.5 | 0.125 | 24.94 |
> > > | | DivOE | 0.25 | 0.125 | 25.36 |
> > > | | DivOE | 0.1 | 0.125 | 24.77 |

---

### Official Review · Reviewer_aWLd · 2023-07-05

**Soundness:** 3 good
**Presentation:** 3 good
**Contribution:** 3 good
**Rating:** 5
**Confidence:** 3

**Summary:**

This paper improves the methods of utilizing auxiliary outlier data for fine-tuning. Specifically, it synthesizes informative outlier samples close to in-distribution at the decision boundary by adding noise to existing auxiliary data and utilizes them in learning. As a result, it shows improved results by applying the proposed method to various outlier exposure methodologies.

**Strengths:**

1. This paper's method is simple and clearly explained. This paper's approach is novel in explicitly adding noise to outlier data to synthesize and utilize outlier data.
2. Theoretically, the proposed method looks a reasonable way to select samples closer to in-distribution data at the decision boundary than existing methods.
3. The experiments are generally fair and show performance improvements when this method is applied as an add-on to a variety of methods. In particular, this paper suggests a method of synthesizing and utilizing outlier data, which presents the possibility of improving the performance of fine-tuning using outlier data.
4. Additionally, this paper shows that this method is an effective way to improve the performance of fine-tuning using outlier data by applying it to a variety of outlier exposure methodologies and showing improved results.

**Weaknesses:**

1. The novelty and superiority in outlier synthesis against DOE [Wang et al., 2023] are not clear against DOE [Wang et al., 2023].
2. The term "diversified" is not well-defined, and it is not clear how it differs from the term "informative" used in ATOM [Chen et al., 2021].
3. The experiments also lack comparison and discussion with the most similar papers MixOE [Zhang et al., 2023] and DOE [Wang et al., 2023] that use outlier synthesis. For example, there is no comparison between the proposed method and DOE [Wang et al., 2023] when it is applied as an add-on to a variety of existing methods.
4. The rationale for using PGD (Projected Gradient Descent) based noise is not well-explained. It is not clear if it has superiority over other attack-based noise.
5. It is not clear how the proposed method of explicitly synthesizing outlier samples differs from implicitly synthesizing them in terms of new effects or novelty.

**Questions:**

1. What is the difference from the most important outlier synthesis methodologies (e.g., MixOE [Zhang et al., 2023] and DOE [Wang et al., 2023]).
2. The fact that the outlier close to the boundary in the left figure of Figure 2 is a diversified outlier is not clearly explained. It is necessary to explain how the informative and diversified are different.
3.	The process of moving from Equation 4 to Equation 5 is not clear. Additional detailed explanations are needed.
4. Equation 4 synthesizes all outlier data and leverages only loss, while Equation 5 synthesizes and calculates loss for a portion of outlier data. It seems that the two equations are different methods. Please explain how the two equations are connected.
5. It is necessary to discuss direct comparison and difference with DOE in TABLE1.
6. Please add the comparison results of each when combined with DOE and MixOE in TABLE2.
7.	Please add experiments on the ImageNet benchmark [A], as discussed in DOE.
8.	Please add experiments on the application of DivOE to OECC [B] in TABLE2.

[A] Tal Ridnik, Emanuel Ben Baruch, Asaf Noy, and Lihi Zelnik. Imagenet-21k pretraining for the masses. In NeurIPS Datasets and Benchmarks, 2021.

[B] Papadopoulos, Aristotelis-Angelos, et al. Outlier exposure with confidence control for out-of-distribution detection. Neurocomputing, 2021, 441: 138-150.


**Limitations:**

Yes. The author adequately addressed the limitations

---

> ### Author Rebuttal · Authors · 2023-08-08
>
> Thank you for your time devoted to reviewing this paper and your constructive suggestions. Here are our detailed replies to your questions.
>
> > **W1, Q1, W5, W3, Q5:** about DivOE, MixOE, DOE
>
> Thank you for the special recommendation of MixOE and DOE for clarity . We would like to explain their differences as follows.
>
> Conceptually, DivOE has a **different underlying motivation** from MixOE and DOE. Our DivOE is proposed for diversifying outlier exposure by extrapolating auxiliary outliers. In comparison, MixOE focuses on enhancing the fine-grained OOD detection by interpolation between ID samples and OOD samples; DOE focuses on improving the generalization of the original outlier exposure* by exploring model-level perturbation.
>
> Technically, DivOE has a **different algorithmic design** from the others. Specifically, DivOE was proposed as a general framework for outlier extrapolation, which bases on OOD samples. It is different from MixOE which directly mix-up ID and OOD samples; It is also different from **DOE that constructs the model perturbation for better optimization**. It is worthy noting **our DivOE is from a different perspective (data-level) compared with DOE (model-level), and they are orthogonal and combinable**. DOE regards regularizing training on the same outliers with different weight perturbations as implicitly synthesized but does not change the outliers. The optimization of DOE is constrained by the original training samples and is also sensitive to tuning the model perturbation.
>
> Empirically, as shown **in Table 1 of the original submission**, we have directly **compared MixOE with our DivOE**. The results show its performance is worse than our DivOE or even the original OE. **For DOE, since it is from a different perspective and orthogonal, we compare DOE mainly in Table 2 of the original submission, and here we present Table 1 in attached PDF for further discussion.** It shows: **(1)** DOE is sensitive to the ID classification task and even performs worse than the original OE in CIFAR-100/SVHN/ImageNet; **(2)** DivOE improves both OE and DOE under the corresponding backbones; **(3)** Even under unfair comparison between DivOE (OE backbone) vs. DOE, DivOE (OE backbone) still achieves the comparable or better performance on ImageNet/SVHN/CIFAR100.  We will add more discussion about the above in our draft.
>
> > **W2, Q2:** "diversified" and "informative"
>
> Thank you for the question.
>
> Conceptually, **"diversified"** is proposed to **characterize one internal property** of auxiliary outliers, while the "informative" in [Chen et al., 2021] is a concept for relative comparison. **The intuition behind "diversified"** is that we **hope the outliers can span more informative areas beyond the original ones**, unlike sampling the most "informative" outliers (i.e., near-boundary OOD samples that are close to ID data).
>
> Intuitively, **Group 1 in Figure 2(a)** indicates the auxiliary outliers which **span more informative areas among those other groups**. In other words, the most diversified Group 1 also covers the areas of other groups. We are sorry for the unclear part and will make the caption of Figure 2 clearer.
>
> Here we also utilize the **Maximum Mean Discrepancy (MMD) to show the difference between "diversified" in our submission and "informative" in Table 2 of attached PDF**. We measure the discrepancy between the sampled/manipulated outliers with the original ones, termed as MMD[new, original], to indicate the "informative"; measure the discrepancy half by half within the sampled/manipulated outliers, termed as MMD[half new, half new], to indicate the "diversified". The results show that DivOE owns lower "informative" due to the partial outlier extrapolation but higher "diversified" on the inner dispersion of the outliers.
>
> Technically, in our DivOE, **“diversified” guides the training to expand the auxiliary outliers to the extrapolated distributions but also consider the original ones.** We re-summarized the results of Figure 4(a) to Table 3 in attached PDF and show that our DivOE with the pursuit of “diversified” (conduct partial extrapolation) performs better than those totally considered “informative” (conduct total extrapolation). The results demonstrate the effectiveness and rationality of DivOE considering partial extrapolation for diversification.
>
> > **W4:** rational of PGD
>
> Note that, **PGD is a replaceable choice in the framework of DivOE**. To better understand the effect of these attack-based noise, **we compare it with other inner-attack methods in Table 4 in the attached PDF** and show that **PGD with the uniform distribution loss is generally the better way to achieve higher performance**. We will include these results and discussion in our draft.
>
> > **Q3,Q4:** Explain Eq.4 and Eq.5
>
> Thank you for the insightful comments. **Eq.(4) corresponding to an expectation formulation** based on the whole population for diversifying outlier exposure, and **Eq.(5) is an instance-level realization** as directly conducting distributional manipulation in Eq.(4) is prohibitively expensive. The deduction from Eq.(4) to Eq.(5) requires to merge the same terms in Eq.(4) first, and then specifically set $\beta=\frac{n-rn}{n}$ with a subsequent rescaling. Finally, we have the formulation that a portion of data to conduct maximization part for diversifying the existing data distribution and the remaining other portion of data corresponds to **learning on the original outliers**. We will add more clear explanation between Eq.(4) and Eq.(5).
>
> > **Q6, Q7, Q8**
>
> Due to the space limit, we place our answer in the general response.

---

> > ### Comment · Reviewer_aWLd · 2023-08-16
> >
> > The authors have addressed my concerns very thoroughly and have provided detailed experiments to support their claims.
> > In the experimental aspects where there was ambiguity, the issues have been resolved, and it has been confirmed that DivOE, while simple, can be applied orthogonally to various algorithms, demonstrating its effectiveness.

---

> > > ### Author Response · Authors · 2023-08-16
> > > **Thanks for your positive response!**
> > >
> > > Thank you very much for the positive feedback after reading our response! We are glad to hear that our response solved your concerns and will make sure to incorporate all suggestions in the revision.

---

> ### Author Response · Authors · 2023-08-14
> **[Invitation to rolling discussion] Need further clarification?**
>
> Dear Reviewer,
>
> Thanks very much for your constructive comments on our work. We have tried our best to address the concerns. Is there any unclear point so that we should/could further clarify?
>
> Best regards.

---

### Author Rebuttal · Authors · 2023-08-08

## General Response

We appreciate all the reviewers for their thoughtful comments and suggestions on our paper.

We are very glad to see that the reviewers find our focused problem is **important** (R2,R3) within the OOD detection research, and the method is **novel** (R1,R2,R3,R4), **theoretical reasonable** (R1,R2) and **simple but adaptable** (R1,R2,R3) to various other techniques, and the experiments are **good**, **comprehensive** and demonstrate the **general effectiveness** of our DivOE (R1,R2,R3,R4). We are also pleased that the reviewers find our writing is **very clear** and **easy to understand** (R2).

We have tried our best to address the reviewers' comments and concerns in **individual responses to each reviewer** with comprehensive experimental justification. The reviews allowed us to improve our draft and **the contents added** in the revised version and **the attached PDF** are summarized below:

**From Reviewer aWLd**

- Clarify and compare the difference of DivOE with MixOE and DOE (see Tables 1,2 in original submission and Table 1 in PDF)
- Explain and compare the difference between "diversified" and "informative" (see Tables 2,3 in PDF)
- Conduct a comparison to explain the rationality of PGD-based attack (see Table 4 in PDF)
- Extend Table 2 with all the add-on varieties (see Table 5 in PDF)
- Conduct experiments on the Large-scaled ImageNet dataset (see Table 6 in PDF)
- Add application of DivOE with OECC (see Table 7 in PDF)

**From Reviewer vD9G**

- Compare the specific outlier numbers affected by each method (see Table 8 in PDF)
- Conduct experiments on the Large-scaled ImageNet dataset (see Table 6 in PDF)
- Show the effect of $\alpha$ in the training/synthetic steps of DivOE (see Table 9 in PDF)
- Incorporating data augmentation methods under the framework of DivOE (see Table 10 in PDF)

**From Reviewer 1Wzd**

- Check the performance when the number of auxiliary outliers is changed (see Table 11 in PDF)
- Discuss and compare performance with GAN-based methods (see Table 12 in PDF)
- Include a comparison of time and memory cost (see Table 13,14 in PDF)
- Conduct experiments on the Large-scaled ImageNet dataset (see Table 6 in PDF)

**From Reviewer Mh63**

- Design additional experiments for the rationality of attack-based inner-maximization (see Tables 4,15 in PDF)
- Check the performance when the number of auxiliary outliers is changed (see Table 11 in PDF)
- Clarify the results considering non-perturbed, perturbed, or both samples in DivOE (see Figure 4(a) in the original submission)
- Clarify the results across five different OOD test sets (see Table 16 in PDF)
- Visualize prediction results of the adversarial perturbation process (see Figure 1 in PDF)

**We appreciate your comments and time!** We have tried our best to address your concerns and revised the paper following the suggestions. **Would you mind checking it and confirming if there are any unclear parts?**

---
### Some rest answers:

For **Reviewer aWLd**

> **Q6,Q7,Q8:** about experiments...

Thanks for the constructive suggestions. **For adding other methods to Table 2**, we conduct extra experiments accordingly and summarize the results **in Table 5 in attached PDF**. The results show that our proposed DivOE can consistently improve performance on the original basis compared with other counterparts across different setups. We will update the complete results in our draft. **For the large-scale experiments**, We accordingly add experiments on the ImageNet benchmark **in Table 6 in attached PDF**, which adopts ImageNet 21k as auxiliary outliers like DOE. The results confirm the effectiveness of our DivOE. **For OECC**, we add the experiments on OECC with our DivOE **in Table 7 in attached PDF**, and will merge them into the original Table 2.

For **Reviewer Mh63**

> **Q5:** about the adversarial perturbation process...

Thanks for the constructive question. From the algorithmic perspective, **the adversarial perturbation with uniform distribution loss is a non-target perturbation**. The intuition is to perturb the OOD sample to be more like ID regarded by the model. Maximizing uniform distribution loss has no requirement for each OOD sample to be perturbed into some specific ID classes. Thus, it is actually **automatically decided by the gradient for the specific perturbation directions**.

To better understand the perturbation effect, we conduct the **case study on the specific example in Figure 1 in attached PDF** for their specific prediction results during the adversarial perturbation process. It can be find that after the adversarial perturbation, **the prediction result of the second example will be concentrate on one specific class with high confidence**. By enlarging the perturbation strength $\epsilon$, the example will be perturbed to be confidently predicted on different classes. We will update more similar visualization to illustrate the movement of the perturbation process on specific samples in our draft.

To provide an overview of the prediction result change, we also check it in our experiments and show in the following table, the results show that after our adversarial perturbation, **most of the OOD samples will be regarded as different ID classes before the perturbation**. It also demonstrates the our extrapolation can finding different outlier distributions beyond the original ones.

| Dataset | Same class | Different class|
| :---: |  :---: |  :---: |
| CIFAR-100 | 4507 | **20517** |
| CIFAR-10  | 9802 | **15222** |

---

### Comment · Area_Chair_xP5r · 2023-08-17
**Author-Reviewer Discussion Period Closing Soon**

Thank you reviewers for your work in evaluating this submission, and thank you authors for responding to the reviewers’ questions and concerns. We are entering the final phase of the discussion period, which will run until August 21st, and some of the authors' responses have to been acknowledged by all the reviewers.

Reviewers: If you have any lingering questions or comments on the rebuttal or the responses, now is the time to express them. At the very least, please acknowledge that you have read the authors’ response to your review.

Thank you everyone for making the review process a fruitful, constructive, and civil process.

AC

---

> ### Author Response · Authors · 2023-08-18
> **Thanks for the Constructive Discussion!**
>
> Dear Area Chair and Reviewers,
>
> We sincerely appreciate the area chair for chairing the discussion process, and would like to thank the reviewers' time and valuable feedback again for the discussion. We are pleased that the reviewers found our rebuttal resolved their concerns and they provided insightful and constructive comments, which we believe will make our claims clearer and significantly enhance the paper's strength. We will incorporate all the suggestions into our manuscript revisions, and welcome any additional discussion during this phase.
>
> Best regards,
>
> Authors of Submission783

---

### Decision · Program_Chairs · 2023-09-21

**Decision:**

Accept (poster)

**Comment:**

This paper focuses on out-of-distribution detection and addresses the weakness of previous outlier exposure-based methods when dealing with a small number of auxiliary outliers. The authors add perturbations to the given outliers to expand the surrogate OOD (out-of-distribution) distributions. The synthesized OOD distributions help bridge the gap between the limited original outliers and in-distribution samples, thereby improving the model's ability to detect OOD samples. During the author-reviewer discussion, most of the previous concerns were addressed, including the comparison with other OE (outlier exposure) works, the method's details, and the sufficiency of the experiments. The reviewers consistently provided positive comments on this paper. If the authors incorporate the suggestions and additional results into the manuscript, this paper could be further improved. Overall, I recommend accepting this paper.